

# A buoyancy, balance and stability challenge to the hypothesis of a semi-aquatic *Spinosaurus* Stromer, 1915 (Dinosauria: Theropoda)

Donald M. Henderson

Royal Tyrrell Museum of Palaeontology, Drumheller, Alberta, Canada

## ABSTRACT

A recent interpretation of the fossil remains of the enigmatic, large predatory dinosaur *Spinosaurus aegyptiacus* Stromer 1915 proposed that it was specially adapted for a semi-aquatic mode of life—a first for any predatory dinosaur. To test some aspects of this suggestion, a three-dimensional, digital model of the animal that incorporates regional density variations, lungs and air sacs was generated, and the flotation potential of the model was investigated using specially written software. It was found that *Spinosaurus* would have been able to float with its head clear of the water surface, although it was laterally unstable and would tend to roll onto its side. Similarly detailed models of another spinosaurid *Baryonyx* (*Suchomimus*) *tenerensis* Sereno et al. 1998, along with models of the more distantly related *Tyrannosaurus rex* Osborn 1905, *Allosaurus fragilis* Marsh 1877, *Struthiomimus altus* Lambe 1902, and *Coelophysis bauri* Cope 1887 were also able to float in positions that enabled the animals to breathe freely, showing that there is nothing exceptional about a floating *Spinosaurus*. Validation of the modelling methods was done with floated models of an alligator and an emperor penguin. The software also showed that the center of mass of *Spinosaurus* was much closer to the hips than previously estimated, similar to that observed in other theropods, implying that this dinosaur would still have been a competent walker on land. With its pneumatised skeleton and a system of air sacs (modelled after birds), the *Spinosaurus* model was found to be unsinkable, even with its lungs deflated by 75%, and this would greatly hinder a semi-aquatic, pursuit predator. The conclusion is that *Spinosaurus* may have been specialized for a shoreline or shallow water mode of life, but would still have been a competent terrestrial animal.

# INTRODUCTION

At the time of their initial discoveries in the 19th century, there were conflicting views about the preferred habitats of dinosaurs. The very largest ones, the sauropods, were claimed by some authors to be capable of a fully terrestrial mode of life (*Mantell, 1850*; *Phillips, 1871*), while others argued for an aquatic one (*Owen, 1875*;

Corresponding author
Donald M. Henderson,
don.henderson@gov.ab.ca

*Hatcher, 1901*). The relatively smaller hadrosaurs, while still impressively big when compared to most modern terrestrial herbivores, were typically thought to be mainly aquatic. A series of anatomical features that were interpreted to be adaptations for an amphibious life were regularly listed for these animals (*Leidy, 1858*; *Cope, 1883*)— webbed hands, deep tails for sculling, etc. In contrast, theropods of all sizes were interpreted as fully terrestrial animals that could not swim. In fact, the aquatic adaptations of hadrosaurs were frequently interpreted as a way to escape predatory theropods by having the former dash to safety in the water, while the latter were left frustrated and hungry on land (*Jackson, 1972*). However, as early as the 1950s it was argued that it was not physically realistic to interpret some dinosaurs as being aquatic, for example sauropods (*Kermack, 1951*). Beginning in the 1960s and 1970s with *Ostrom's (1964)* re-interpretation of hadrosaurs as fully terrestrial animals, and *Bakker's (1971)* arguing for terrestrial sauropods, the interpretation of all dinosaurs as fully terrestrial animals was starting to take hold. During the past 47 years, as our knowledge of dinosaurs has increased exponentially (*Wang & Dodson, 2006*), this 'terrestrialization' of dinosaurs has seemed unshakeable.

The idea that spinosaurids might have been piscivorous appears to have begun with *Taquet (1984)*. Since then there have been suggestions that *Spinosaurus* and its close relatives might have had a strong association with aquatic environments. *Charig & Milner (1997)* accepted the idea of the new english spinosaurid *Baryonyx walkeri* as a fish eater, but preferred to keep the animal on shore. From an analysis of calcium isotopes in vertebrate teeth from mid-Cretaceous continental biotas of North Africa, *Hassler et al. (2018)* found that spinosaurids had a strong freshwater food source signal. Additionally, *Amiot et al. (2010)*, based on analyses of oxygen isotope ratios ($\delta^{18}O_p$) from biogenic apatites from a wide range of spinosaurid remains, proposed that spinosaurids spent extended periods in freshwater. They also suggested that they may have fed on both terrestrial and aquatic prey. Despite these suggestions, they did include the following statement in their paper 'However, their [spinosaurid] postcranial anatomy differs relatively little from that of usual, large bipedal theropods, and is not particularly suggestive of aquatic habits.' (*Amiot et al., 2010*, p. 139).

Based on a skeletal reconstruction derived from one partial, associated skeleton and several isolated, partial specimens from other localities of the Late Cretaceous dinosaur *Spinosaurus aegyptiacus* (*Stromer, 1915*), and a functional interpretation of the resulting body form, along with anatomical details, *Ibrahim et al. (2014)* made a case for this exceptionally long and 'sail-finned' dinosaur being a semi-aquatic predator, and particularly well-adapted for pursing prey in the ancient rivers recorded by the Kem Kem beds rocks exposed in Morocco. This interpretation of an extinct theropod as being semi-aquatic was much more forcefully stated than previous suggestions, and generated much media attention (*Tarlach, 2014*; *Coghlan, 2014*).

Following after the article of *Ibrahim et al. (2014)*, other authors took up the idea of *Spinosaurus* as a piscivore, or even as an active aquatic predator. *Vullo, Allain & Cavin (2016)* outlined the convergence in the shapes of the margins of the jaws and of the teeth of *Spinosaurus* and that of the predatory pike conger eels (members of the

family Muraenesocidae). These authors cautiously suggested that spinosaurs would have been well adapted to forage in aquatic settings like the eels, but did not say anything about semi-aquatic habits for spinosaurids. A very speculative paper on the swimming abilities of *Spinosaurus* and the function of the dorsal 'sail' by *Gimsa, Sleigh & Gimsa (2016)* employed qualitative comparisons between crocodilians, large, predatory fishes (both chondrichthyan and osteichthyan) and *Spinosaurus*. These authors envisaged *Spinosaurus* as an animal capable of becoming fully immersed and employing lateral undulation in the pursuit of prey. These authors also hoped that more quantitative studies in the form of hydrodynamical and biomechanical analyses would refine our understanding of the functions of the peculiar anatomy spinosaurids.

The gross morphological features of extinct dinosaurs do not immediately suggest any capacity for a mode of life that had an aquatic component. Their dorsal, and often their caudal vertebrae as well, were tightly articulated with little capacity for lateral motion that could assist with aquatic locomotion via lateral undulation. In particular, the theropod clade Tetanura (*sensu Gauthier, 1986*) with their stiffened tails, would have been most unlikely to have been tail-propelled. Spinosaurids belong to the latter clade (*Carrano, Benson & Sampson, 2012*). The parasagittal hind limbs of all dinosaurs, being held in place with the head of the femur deeply implanted in the acetabulum, would also seem unlikely to have performed well in an aquatic setting. Modern, semi-aquatic crocodilians evolved from thoroughly terrestrial animals, and show changes in their spines and hips, especially their capacity to switch the hindlimb orientation between a high walk and a semi-sprawl, that make them much better adapted to a semi-aquatic life (*Grigg & Kirshner, 2015*). There are examples from around the world of dinosaur fossils recovered from marine settings: hadrosaurs—*Eotrachodon orientalis Prieto-Márquez, Erickson & Ebersole 2016*; theropods—*Scipionyx samniticus Dal Sasso & Signore 1998* and *Nothronychus mckinleyi Kirkland & Wolfe 2001*; ankylosaurs—*Kunbarrasaurus ieversi Leahey et al. 2015*. However, these examples are all interpreted as thoroughly terrestrial animals that got washed out to sea.

The emphatic claim by *Ibrahim et al. (2014)* of a semi-aquatic theropod dinosaur inspired further investigation of the aquatic potential of *Spinosaurus,* and some specially written software was used to test the center of mass (CM), buoyancy and equilibrium of an immersed digital model of the animal. To put the results from an analysis of an immersed *Spinosaurus* into context, the floating capabilities of five other theropods, including another spinosaurid were also tested. The collective body masses of these five animals span almost four orders of magnitude, allowing for the investigation of the effects of body size on the potential for flotation and stability of immersed theropods.

## MATERIALS AND METHODS

The digital *Spinosaurus* model used in the current study was based on the illustration provided in Fig. S3 of the Supplementary Materials of *Ibrahim et al. (2014)*, and the geometry of the model was taken from this figure using the slicing method of *Henderson (1999)*. The length of the model was also based on the new restoration of *Spinosaurus* by *Ibrahim et al. (2014)*. These authors state that a life size replica of

*Spinosaurus*, generated from their new skeletal data, was 'over 15 m in length' (last sentence, third paragraph). As measured from the tip of its snout to the tip of its tail, the length of the present digital model is 15.55 m. The illustration in the supplementary materials of *Ibrahim et al. (2014)* shows the head tipped forward and the jaws agape, but for the digital model the mouth was closed by rotation of the contour of the mandible about the illustrated quadrate-articular joint, and the head elevated via rotations of the slices defining the neck until the occlusal plane of the mouth was horizontal. Although dorsal views of the skull of the reconstructed skull *Spinosaurus* are available, a dorsal view of new whole body reconstruction was not. The relative transverse dimensions of the body posterior to the head were guided by reconstructions showing dorsal views of other large dinosaurs by palaeoartists, for example Greg *Paul (1988)*. The new restoration of *Spinosaurus* by *Ibrahim et al. (2014)* is a composite derived from several specimens, and there will always be a level of uncertainty as to the actual dimensions and relative proportions of the various body regions. As the claims for a semi-aquatic mode of life for *Spinosaurus* were associated with this new restoration, it was the one used for model generation and buoyancy/stability testing.

Five other theropods, four of which were not closely related to each other or to *Spinosaurus*, were chosen for comparison with the latter. These were *Coelophysis bauri* (Ceratosauria), *Struthiomimus altus* (Ornithomimosauria), *Allosaurus fragilis* (Carnosauria), *Tyrannosaurus rex* (Tyrannosauridae), and the spinosaurid *Baryonyx* (*Suchomimus*) *tenerensis* (Fig. 1). The illustrations used as sources for the models are listed in Table 1. It has been suggested that the fossil remains of *Suchomimus* are not distinct enough from *Baryonyx* to merit the erection of a new genus (*Holtz, 2012*; *Sues et al., 2002*), and this suggestion is followed here. The two criteria governing these choices of theropod for comparative purposes were that the animals be known from enough skeletal material to produce reliable, whole body reconstructions, and that they span a range of body sizes to enable investigation of the effects of body size on the ability of theropods to float. There are allometric changes in body shapes as theropods increase in size over time, with the trunk region becoming deeper, broader and relatively shorter, and the hind limbs becoming more massive (*Henderson & Snively, 2004*). It was felt important to check if these changes in body proportions would affect the ability of the animals to float.

For the other models, the axial body and limb shapes used in their construction were obtained using the same three-dimensional, mathematical slicing method of *Henderson (1999)*. The basic axial body tissue density of the models was set to be the same as that of water—1,000 gm/l. However, this was modified in certain regions to reflect aspects of theropod anatomy. The system of air sacs within the bodies of extant birds represents about 15% of their axial body volume (*Proctor & Lynch, 1993*), and this observation was used to adjust the basic axial body densities of the models. From fossil evidence of extensive pneumatisation of the skeletons of extinct theropod dinosaurs, and the inference that these animals had a system of air sacs similar to those of extant birds (*O'Connor & Claessens, 2005*), the pre-caudal, axial densities of the models were reduced by 15% to 850 gm/l to incorporate the density reductions associated with the presumed air sacs in the hips, trunk, and neck. Lacking evidence for differences in the

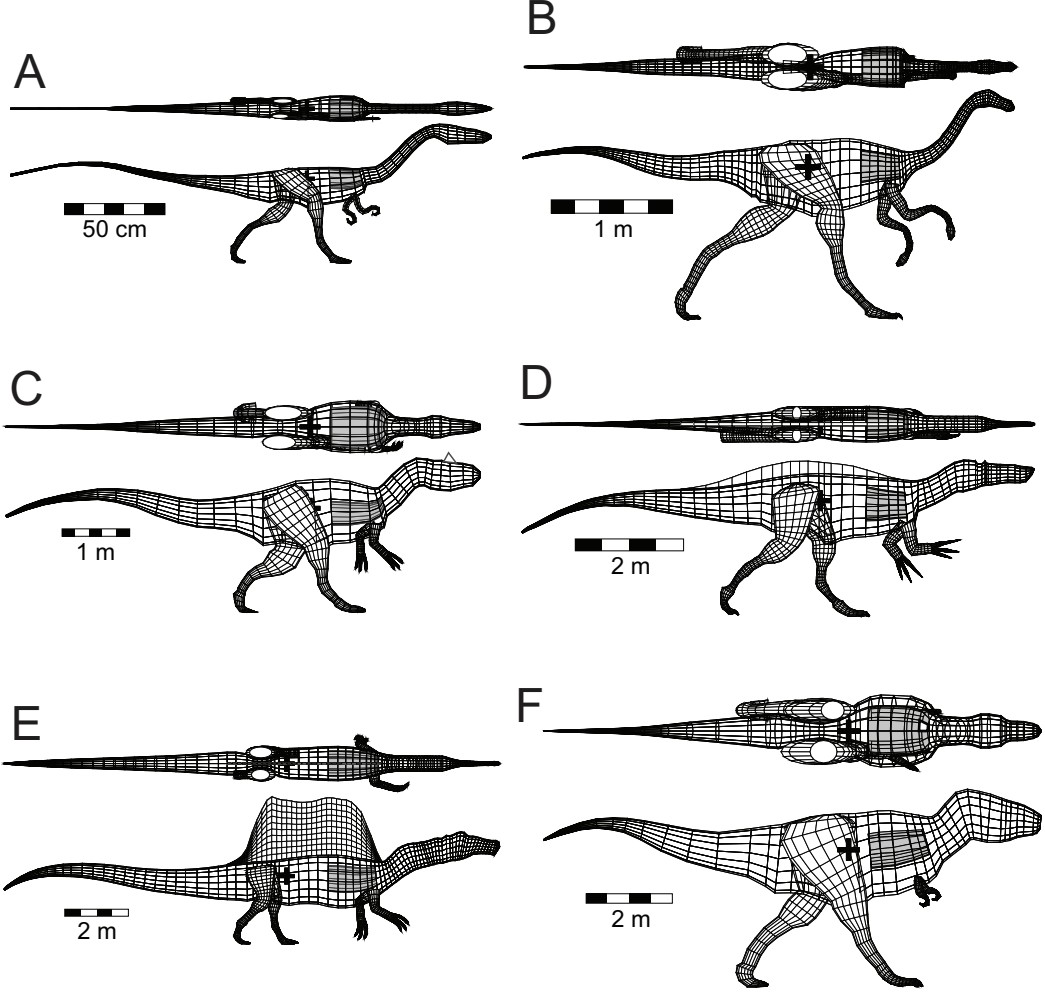

**Figure 1 Dorsal and lateral views of the theropod models used for flotation tests.** (A) *Coelophysis bauri*; (B) *Struthiomimus altus*; (C) *Allosaurus fragilis*; (D) *Baryonyx* (*Suchomimus*) *tenerensis*; (E) *Spinosaurus aegyptiacus*; (F) *Tyrannosaurus rex*. Animals in order of increasing mass. Lung volumes and positions are represented by the dark grey cylinders in the chest regions. Black '+' denotes the computed center of mass. See Tables 1 and 2 for model image sources and model details, respectively.

**Table 1 Sources of illustrations used to generate the theropod and alligator body forms.**

| Taxon | Image sources |
|---|---|
| *Alligator mississippiensis* | *Neill (1971)* |
| *Coelophysis bauri* | *Paul (1988)* and *Currie (1997)* |
| *Struthiomimus altus* | *Paul (1988)* |
| *Allosaurus fragilis* | *Paul (1988)* |
| *Baryonyx (Suchomimus) tenerensis* | *Sereno et al. (1998)* and Hartman (in *Holtz, 2012*) |
| *Spinosaurus aegyptiacus* | *Ibrahim et al. (2014)* |
| *Tyrannosaurus rex* | *Paul (1988)* and *Currie (1997)* |

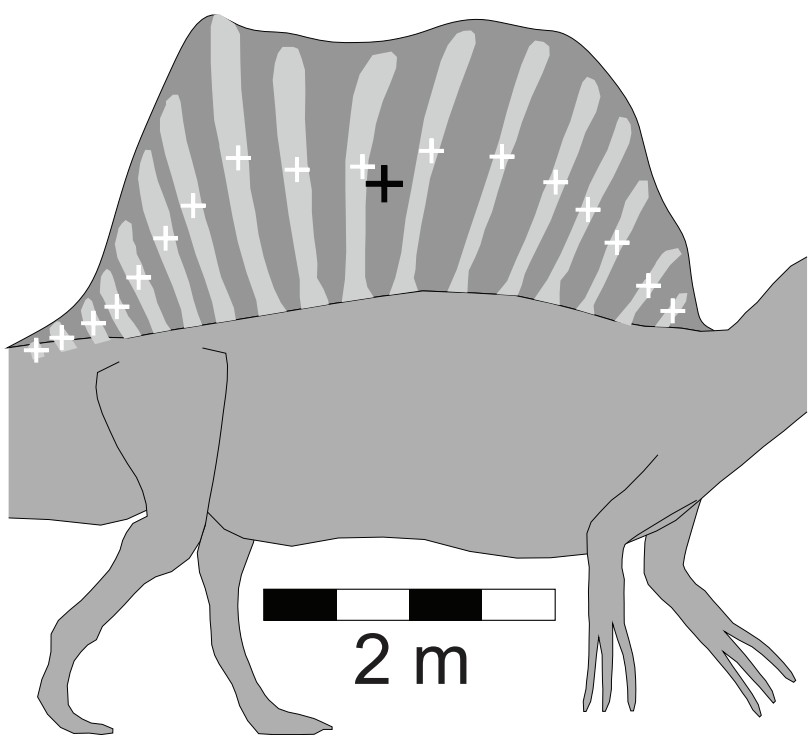

**Figure 2 Detailed view of the *Spinosaurus* 'sail' and its associated neural spines (after *Ibrahim et al. (2014)*).** These details were used to determine the relative fractions of the bony and soft tissue components of the sail which were then used to compute the mass and center of mass of the sail. These latter two values were components in the final calculations of the mass, center of mass, and buoyant characteristics of the complete *Spinosaurus* model. Small white '+'s are the centroids of the individual spines. Large black '+' is the centroid of the entire sail. See 'Methods' for details of the calculations.

sizes and relative proportions of air sacs in extinct theropods, the most parsimonius assumption is that they were all of similar construction and proportions. The presence of pneumatised bones in theropod skulls, along with the nasal and oral cavities, led to the same reduced density value being assumed for the heads. A lung cavity was also produced for each model and located in the antero-dorsal portion of the thorax. For all the models the lung volume was set at approximately 9% of the axial body volume based on observations of living reptiles (*Gans & Clark, 1976*). The theropods used in the present study are assumed to have been non-flying, so the use of a lung volume scaling seen in living birds (*Schmidt-Nielsen, 1989*, Table 9.2) was not considered appropriate. The mass deficits represented by the lungs were incorporated into the determination of the buoyant states of the models. Lastly, the limbs with their substantial bone component were assigned a slightly higher density of 1,050 gm/l. The masses and CM of all the models were estimated with the method presented in *Henderson (1999)*.

Among the distinctive features of *Spinosaurus* is the large dorsal 'sail' (Fig. 1E). Given the size and position of the sail, and its potential to affect the equilibrium of a floating *Spinosaurus*, special attention was given to its construction and mass estimation, and this was guided by the comments on the sail by *Ibrahim et al. (2014)*. Figure 2 presents details

**Table 2 Body lengths, total mass, and component masses for the eight models used in the present study.**

| | Alligator mississippiensis | Aptenodytes forsteri | Allosaurus fragilis | Baryonyx (Suchomimus) tenerensis | Coelophysis bauri | Spinosaurus aegyptiacus | Struthiomimus altus | Tyrannosaurus rex |
|---|---|---|---|---|---|---|---|---|
| Length (m) | 3.07 | 1.25 | 7.35 | 9.78 | 2.52 | 16.0 | 4.35 | 12.0 |
| Total mass (kg) | 133 | 46.3 | 963 | $2.14 \times 10^3$ | 10.3 | 6,500 | 201 | 9,750 |
| Mean body density (kg/m$^3$) | 952 | 968 | 818 | 840 | 828 | 833 | 858 | 851 |
| Axial mass (kg)[1] | 106 | 44.2 | 757 | $1.29 \times 10^3$ | 7.77 | 5,470 | 119 | 6,030 |
| Single arm mass (kg) | 1.58 | 0.354 | 7.12 | 20.0 | 0.0413 | 54.0 | 3.67 | 10.3 |
| Single leg mass (kg) | 4.88 | 0.704 | 121 | 216 | 1.20 | 295 | 40.7 | 1,430 |
| Lung volume (l) (% Axial volume) | 11.4 (9.10) | 1.05 (23.5) | 97.8 (9.98) | 149 (9.09) | 1.08 (10.8) | 662 (10.0) | 14.5 (9.53) | 837 (10.5) |
| CM (x, y)[2] | (1.86, −0.146) | (0.539, −0.118) | (4.50, 0.645) | (5.50, 0.814) | (1.48, 0.148) | (8.85, 1.00) | (2.35, 0.416) | (7.01, 1.35) |
| Horizontal relative CM (%)[3] | 27.7 | 71.6 | 19.2 | 19.0 | 27.2 | 20.9 | 15.3 | 28.6 |

**Notes:**
Listed alphabetically by genus from left to right.
[1] Axial mass reduced by an equivalent mass of water represented by the lung cavity and excludes the mass of the sail for *Spinosaurus*.
[2] Centre of mass: horizontal position expressed as meters from the tip of the tail, vertical position is meters above lowest point of the axial body. For *Alligator* and *Aptenodytes* vertical CM is from floating models and measured relative to water surface.
[3] Horizontal relative CM: distance in front of acetabulum expressed as a percentage of the gleno-acetabular distance.

of the sail relevant to the construction of its model. Digitizing the outline of the entire sail, and computing its lateral area by the triangular decomposition method outlined in *Henderson (2003a)*, gives a value of 6.60 m$^2$. Digitizing the perimeters of the neural spines associated with the reconstruction of the sail shown in *Ibrahim et al. (2014*, fig. 2*)*, and computing their net area, reveals that the combined lateral areas of these bones, 2.45 m$^2$, is equivalent to slightly more than one-third of the lateral area of the entire sail. The volume of bone comprising the sail is given by the product of the lateral area of the neural spines multiplied by an assumed transverse thickness of 2.25 cm, giving a value of 0.0550 m$^3$. Lacking information to the contrary, the sail was assumed to be covered with skin to a depth of one cm on both sides, giving a total thickness of 4.25 cm. The total volume of the sail is the product of its full lateral area, 6.60 m$^2$, and its estimated maximum thickness, and this gives a value of 0.281 m$^3$. Subtracting the volume of the bony component of the sail from the total sail volume gives a volume measure for the soft tissue component. The soft and bony tissues of the sail were assumed to have densities of 1,000 and 2,000 gm/l, respectively. With the above volume and density values for the soft and hard components of the sail, the total mass of the sail was estimated to be 335 kg. The centroid of the sail was computed during the estimation of its lateral area (*Henderson, 2003a*), and taken to be the CM of the sail. The mass of the sail represents approximately 7.5% of the axial body mass, and almost 80% of the mass deficit represented by the lung cavity (Table 2). Assuming a density of 1,000 gm/l, the mass of the one cm

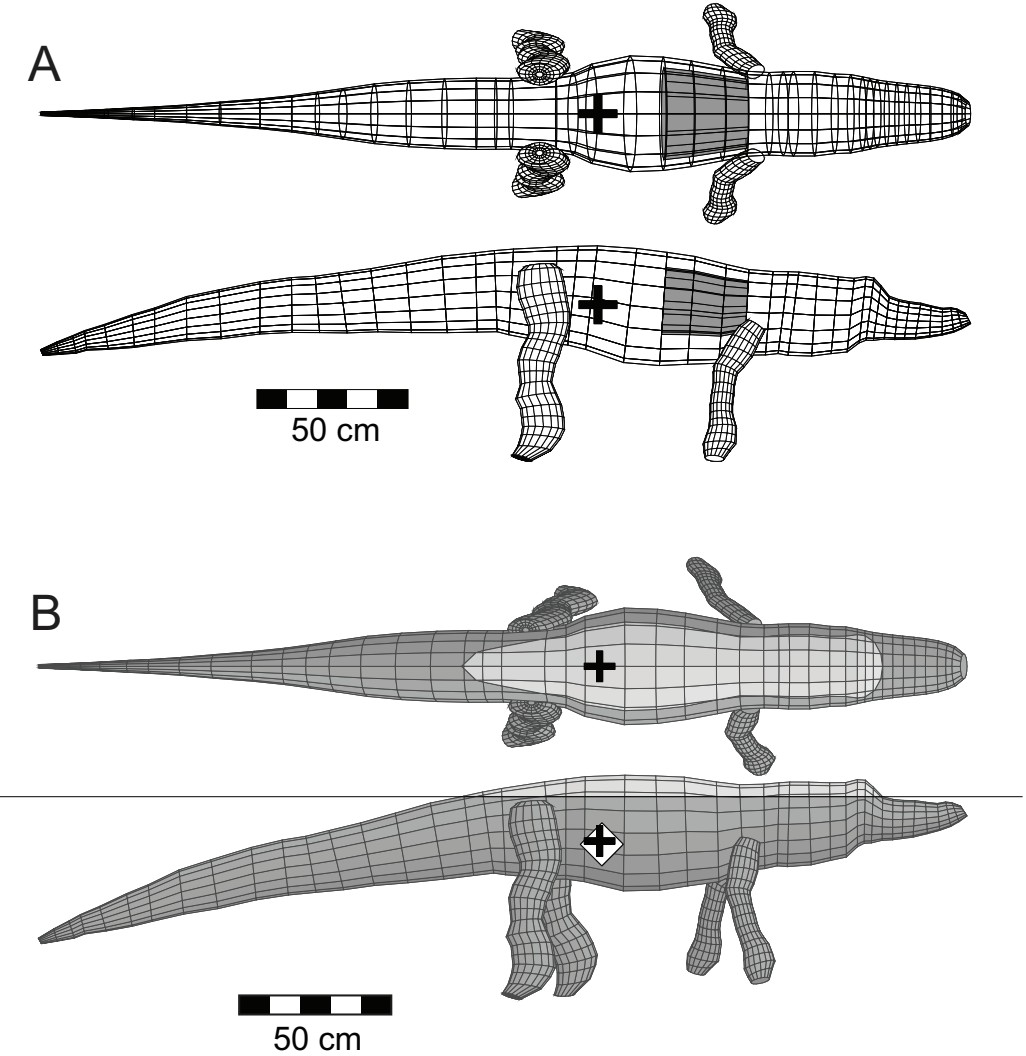

**Figure 3 Three-dimensional alligator (*Alligator mississippiensis*) model as a validation of the methods.** (A) Basic model; (B) floating model that has attained buoyant equilibrium with a fully inflated lung. Thin, horizontal black line is the water surface. Light coloured dorsal regions are 'dry' and exposed to the air. Black '+' denotes the center of mass, while the white '◇' indicates the center of buoyancy. These figures are derived and updated from *Henderson (2003b)*. See Tables 1 and 2 for details of the model and its floating state.

thick layer of skin on one side of the sail is 66 kg. Doubling the thickness of the skin on both sides would increase the sail mass fraction to approximately 8.5% of the axial body mass. The mass and CM of the sail were considered necessary components to ensure an accurate determination of the floating state of *Spinosaurus*.

The mathematical and computational methods used to simulate the immersion of a model tetrapod, and the analysis of a model's floating characteristics, were developed in *Henderson (2003b)*. To ensure that the modelling and the software can replicate the orientation and depth of immersion of a large reptile that can be observed floating today, the software was tested using a model of the semi-aquatic American alligator (*Alligator mississippiensis Daudin 1802*) (*Henderson, 2003b*) (Fig. 3). Crocodylians

share a common ancestry with theropods as both are members of Archosauria. Additionally, the alligator has an elongated body with a substantial, muscular tail similar to that inferred for theropods. As was done with the theropod models, a lung volume equal to 9% of the axial body volume was generated for the alligator model. Unlike what was done for the theropods, the axial body and limb densities were maintained at 1,050 gm/l as crocodilians lack the system of air sacs inferred for theropods.

It was suggested by a reviewer that a test of the software and methods should also be done with a living, aquatic, predatory theropod, that is a diving bird, to see how it would compare to *Spinosaurus*. This was done with a model of an emperor penguin (*Aptenodytes forsteri* Gray 1844). The model was derived from frontal and lateral views of an adult using the 3D-slicing technique, and included both the hind and fore limbs. The total body length from the tip of beak to the tip of the tail was 1.25 m. The post-cervical axial body density was set to 1,000 gm/l, while the neck and head were set to 800 gm/l. Penguins do not have the system of air sacs found in other birds and have denser bones (*Simpson, 1976*), hence the higher axial body density. The limb densities were set to 1,050 gm/l. A lung volume was generated using the bird lung scaling relationships of *Schmidt-Nielsen (1989)*.

It was suggested by another reviewer to test the lateral stability of the floating *Spinosaurus* model, and it was decided to do the same test on the alligator model as well. The traditional naval architecture parameter of the metacentric height (MC) (*Comstock, 1967*) was computed for the full body models and this required two additional parameters to be extracted from the models. The first is the water plane for a model, and this was taken as the area representing the intersection of the floating model with the water surface. As lateral stability is the topic of interest, the second moment of area of the water plane was computed with respect to the longitudinal ($X$) axis located in the sagittal plane. The second parameter is the volume of the immersed portion of the body, and this was extracted from a model's geometry by noting the degree of immersion of each of the sets of cylindrical disks forming the axial body and limbs. The MC, is usually defined as the distance above the keel of a boat, but in the present situation it was taken as the distance below the water surface at the longitudinal position of the CM. MC was computed with the following expression:

$$MC = CB + \frac{I_x}{V} \tag{1}$$

where centers of buoyancy (CB) is the distance of the center of buoyancy from the ventral surface, $I_x$ second moment of area of the water plane, and $V$ is the volume of the immersed portion.

To provide a more intuitive and visual assessment of the lateral stability of the models, another test of the lateral stability of the alligator and *Spinosaurus* models was done. This involved testing the stability of two-dimensional disks representing the cross-sections of the axial bodies of the two models, and presenting the results as selected frames of an animation sequence to show the behaviour of the disks when perturbed. The combined volumes of the limbs of *Alligator* and *Spinosaurus* represent 9.00% and 11.7%, respectively,
of the total volumes of the models (the dorsal sail of *Spinosaurus* was included in its volume measure). As a first approximation, the small contributions to total body mass and buoyancy by the limbs were ignored when performing this stability test. An elliptical disk representing the average cross-section of the axial body of a model was produced by computing the average dorso-ventral and medio-lateral radii from the two slices defining the axial body immediately posterior and anterior to the longitudinal position of the CM in the floating model. This elliptical shape was done as a 'super-ellipse' where the exponent was 2.5 instead of the usual 2. This produces cross-sections of slightly flatter tops, bottoms, and sides than a normal ellipse, and is more biologically plausible than a regular ellipse (*Motani, 2001*). The disk represents a transverse section of the floating axial body, and the competing forces of gravity and buoyancy were assumed to act in the plane of the disk. The mass of the disk is the product of its area, thickness and density, with the value of the latter being the mean density of the whole model with a full lung. As it is of uniform density, the CM of the test disk was taken to be its centroid. An iterative process of analysis involved determining the degree of immersion of the slice to compute the magnitude of the upwards buoyant force and the two-dimensional location of the center of buoyancy. The positively directed buoyant force was added to the unchanging negatively directed weight force, and if the result was positive the disk was moved up by an amount proportional to the magnitude of the difference. Conversely, if the result was negative, the disk was moved downwards. Any horizontal separation between the CB and gravity represented a moment arm for the buoyant force and would produce a turning moment on the disk acting about the CM. After adjusting the vertical position and angular orientation of the disk, the process of testing and shifting was repeated. The disk was considered to be in a final, stable equilibrium state when the difference between the gravity and buoyant forces was less than 1% of the weight force and the torque acting on the disk was less than 0.5% of a predefined reference torque. See *Henderson (2003b)* for more complete details on bringing a floating model to equilibrium.

For the present study, all but one of the flotation simulations were done with the assumption that the models were in freshwater with a density of 1,000 gm/l. The only exception was with the penguin which was floated in seawater with a density of 1,026 gm/l.

The potential effects of increased bone density on the mass and overall density of a floating theropod were checked using three-dimensional, digital models of the non-pedal bones of the hindlimb of *A. fragilis*. Hind limb bones were chosen for this test as the increased density of those of *Spinosaurus* were explicitly mentioned by *Ibrahim et al. (2014)*. The bone geometries were taken from illustrations of the femur, tibia, fibula and metatarsals of *Madsen (1976)*, and their digital models were generated using the methods of *Henderson (1999)*. These bones were analysed in association with the three-dimensional mesh representing the muscles and fleshed out hind limb of the *Allosaurus* model of Fig. 1.

## RESULTS

The whole body and component masses computed for the six theropod models, the alligator and the penguin are presented in Table 2. The striding, non-floating theropod

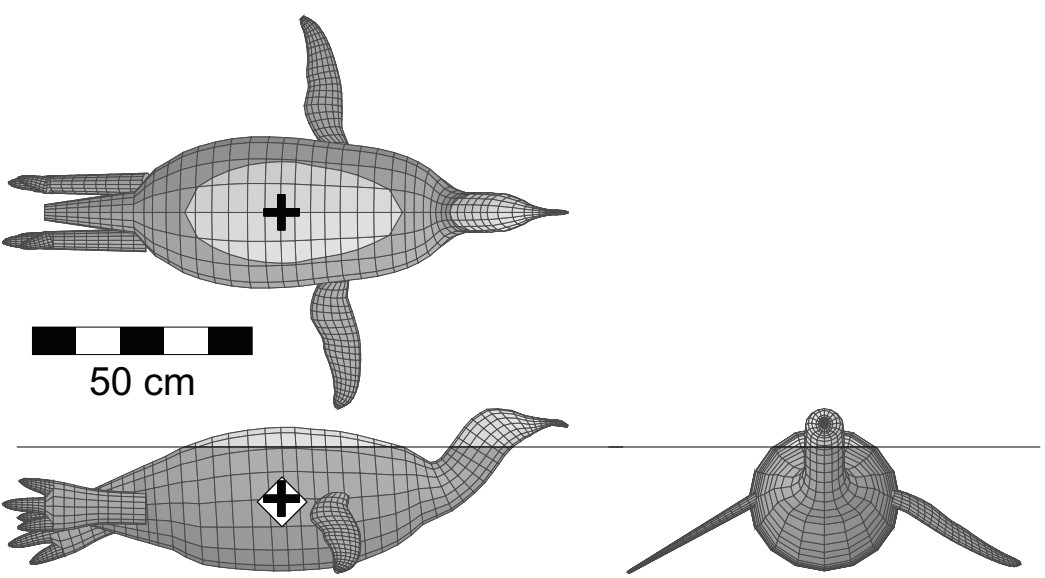

**Figure 4 Dorsal, lateral and anterior views of the floating model of the emperor penguin (*Aptenodytes forsteri*).** This example of an extant, aquatic, predatory theropod was done as another test of the validity of the methods employed with the extinct theropods. The model is in its final, equilibrium flotation state with a full lung, and replicates the situation seen in living emperor penguins floating at the water surface. Unlike all the other flotation tests, this one is done with seawater of density 1,026 gm/l. Colours and symbols as per Fig. 3. See Table 2 for details of the model and its floating state.

models of Fig. 1 all show their CM located just ahead of the hip sockets, but above and between the leading and trailing feet, demonstrating that the animals are balanced with no tendency to tip forward or back. This same is true for the new restoration of *Spinosaurus* by *Ibrahim et al. (2014)*. Even with its rather short legs, the CM of *Spinosaurus* is still positioned above the leading foot, showing that with appropriate stride lengths, this animal could still walk on land (*Gatesy, Baker & Hutchinson, 2009*).

The upper pair of images of Fig. 3 presents the basic mesh form of the alligator model in dorsal and lateral views, together with a grey cylinder indicating the size and position of the lung cavity. The estimated total mass of the 3.07 m long model is 122 kg, and these values are similar to those observed for a 2.89 m female alligator that weighed 129 kg (*Woodward, White & Linda, 1995*). Further demonstrations of the validity of the alligator model and its computed parameters can be found in *Henderson (2003b)*. The lower pair of images of Fig. 3 show the model in stable, floating equilibrium with a fully inflated lung. This final state closely replicates the observed resting positions of both crocodiles and alligators when resting at the water surface (*Grigg & Kirshner, 2015*, Chapter 4). With the model of a floating alligator successfully replicating aspects of a living one, this provides a level of confidence for what is predicted for the extinct theropods.

Figure 4 shows the penguin model with a full lung floating in seawater with a density of 1,026 gm/l, and exhibiting stable equilibrium at the surface. The mean body density of the model was 968 gm/l, and the computed total body mass was 46.3 kg. The average mass of a male emperor penguin is 38 kg (*Dunning, 2008*). The current model is 1.25 m long from the tip of the tail to the tip of the beak, and is larger than

the typical 1.20 m height observed for large males (*Gooders, 1975*). Assuming isometric scaling for the fully mature, adult penguin in this instance, with its body mass being proportional to the cube of body length, the body length ought to be reduced by eight cm to 1.17 m to get the model mass down to the average of 38 kg. The model body orientation and depth of immersion match observations of living emperor penguins at the surface (*Kooyman et al., 1971*), and provides another indication of the reliability of the modelling process. Deflating the model penguin lung by 90% resulted in a mean body density of 989 gm/l, which is still not high enough to make the model negatively buoyant and enable sinking. However, emperor penguins have been observed to inhale prior to diving (*Kooyman et al., 1971*), so the lung deflation test is not particularly relevant. With their highly derived wings and powerful pectoral muscles, penguins are able to overcome the positive buoyancy associated with a full lung and propel themselves downwards underwater (*Lovvorn, 2001*).

For the present study, a criterion for judging whether a normally terrestrial animal was unlikely to drown and could maintain a stable body orientation while immersed was that the head, and the nostrils in particular, were clear of the water surface so that the animal could see and breathe. Figure 5 presents the final, equilibrium floating states for the two spinosaurid models with full lungs. In each case, the models float with their heads and nostrils above the water, and their CM and buoyancy are nearly coincident. As postulated by *Ibrahim et al. (2014)*, the sail of *Spinosaurus* does stay visible while the animal is floating. The orientations of the heads and necks of these models were not altered from the basic, 'terrestrial' versions shown in Fig. 1. The mass of the low crest associated with the *Baryonyx* (*Suchomimus*) model represents 2.2% of the axial body mass. This smaller mass, when compared to the larger 7.5% relative mass of the *Spinosaurus* sail, and combined with the fact the center of the crest lies close to the CM of the whole body, leads to the position and mass of the *Baryonyx* (*Suchomimus*) crest having only a very minor effect on the overall, final CM of the model.

Figure 6 presents the floating equilibrium states of the four other comparative theropod models. The first thing to notice is that all four animals/models can float, and that their heads are clear of the water surface. The heads of the *Coelophysis* and *Tyrannosaurus* models needed to be dorsiflexed by 20° and 15°, respectively, to elevate them enough so that the tips of their snouts (nostrils) were above the water surface. These head elevations were done via a series small increments applied to the each of the model slices defining the necks, until the sum of the rotations applied to individual slices equalled the required total head lifting angle. An additional feature is that the floating states appear to be independent of body size, with the same proportions of the bodies being exposed above the water line. The only apparent difference is that the *Coelophysis* model floats with body tipped much more forward, when compared to the others. This may be related to two aspects of the body shape of *Coelophysis*. The much more attenuated, and slender axial body, with less of the body mass concentrated about the hips, and the much longer neck, which will not only represent a larger fraction of the total body mass, but in combination with the head, will also exert a stronger turning moment on the body.

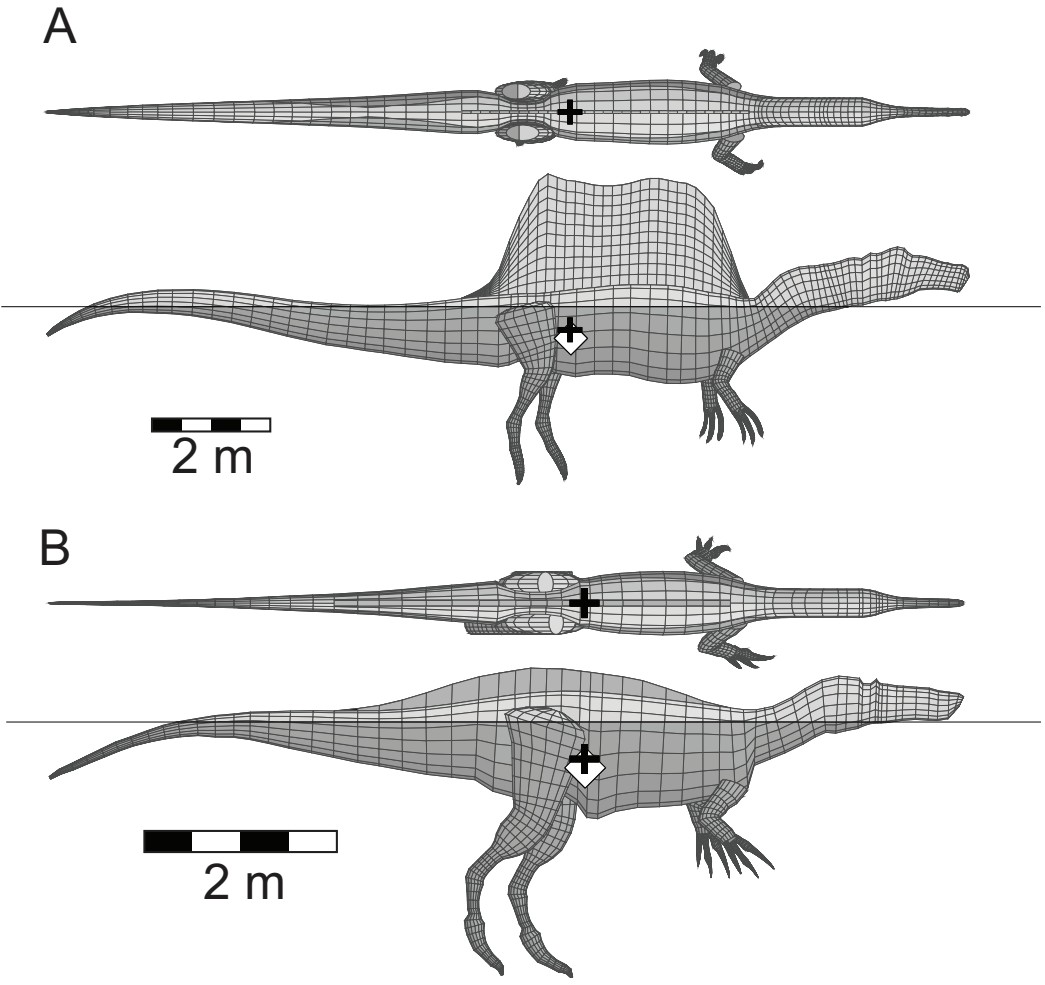

**Figure 5 Floating spinosaurids in lateral and dorsal views.** (A) *Spinosaurus aegyptiacus*; (B) *Baryonyx* (*Suchomimus*) *tenerensis*. Determination of the buoyant state required knowing the masses and centers of mass of the axial body (accounting the presence of a lung), all four limbs, and in both cases, the dorsal 'sail.' See Table 2 for model details.

Figure 7 presents graphically the locations of the CB, CM, and the MC of the alligator and *Spinosaurus* models. These three quantities in the alligator are all virtually coincident with one another, with just millimeters separating them. A MC located below the CM of an immersed object indicates an unstable situation. The position of the MC of the alligator is computed as being almost identical to that of the CM, the separation being less than one mm, and given the asymptotic nature of the approach to equilibrium, these quantities can be considered fully coincident. The closeness of the three quantities indicates that any moment arms associated with misaligned buoyant and gravitational forces will be extremely small. In complete contrast, the positions of CB, MC, and CM of the *Spinosaurus* model clearly demonstrate an unstable situation, with the center of gravity located 22 cm above the MC.

Figure 8 shows the results of the two-dimensional-disk lateral stability test conducted for the alligator model with the disk representing the transverse section of the body at

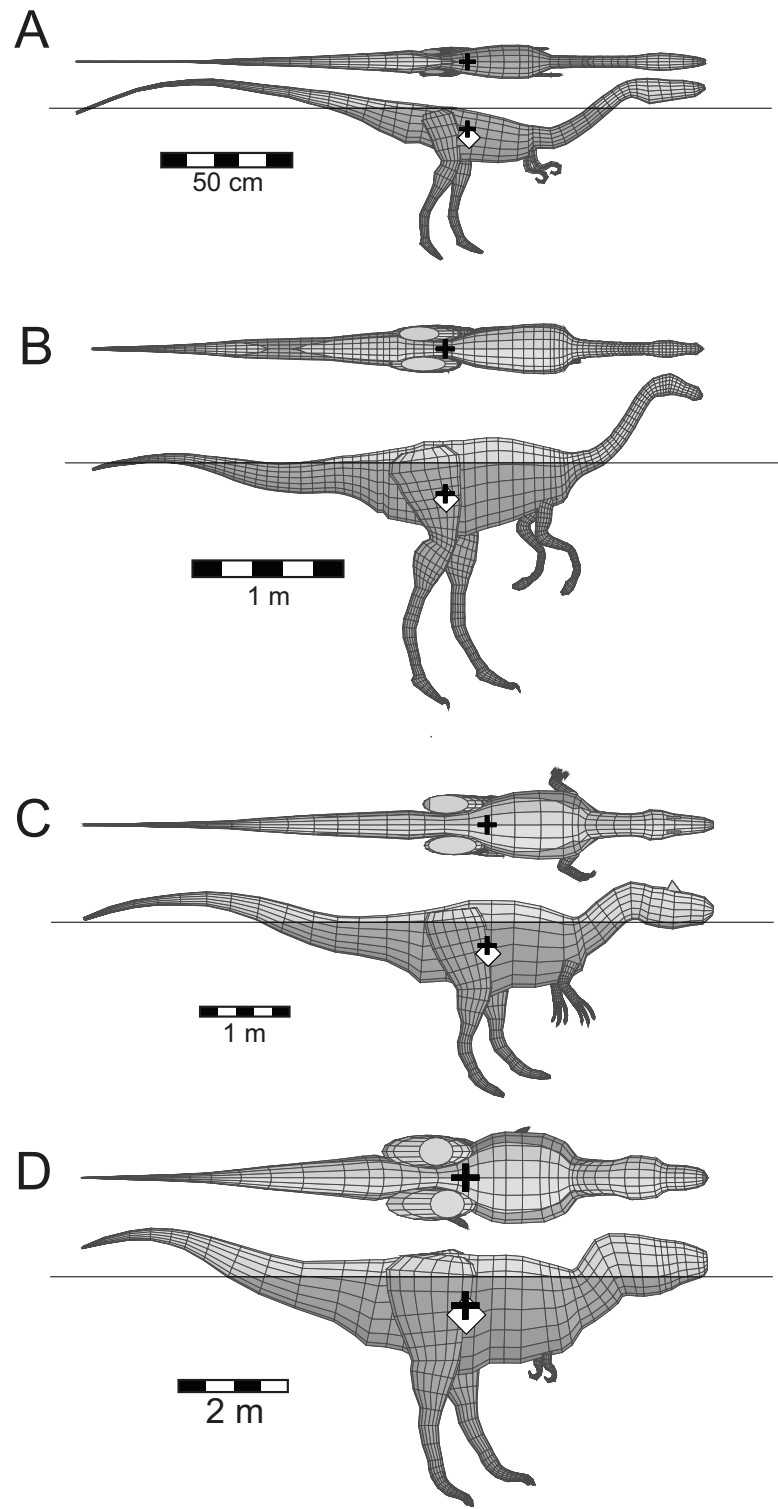

**Figure 6 Floating theropods with masses ranging from 10.3 to 9,360 kg.** (A) *C. bauri*; (B) *S. altus*; (C) *A. fragilis*; (D) *T. rex*. See Fig. 3 explanation of symbols. All models floated with full lungs. See Table 2 for model details.               

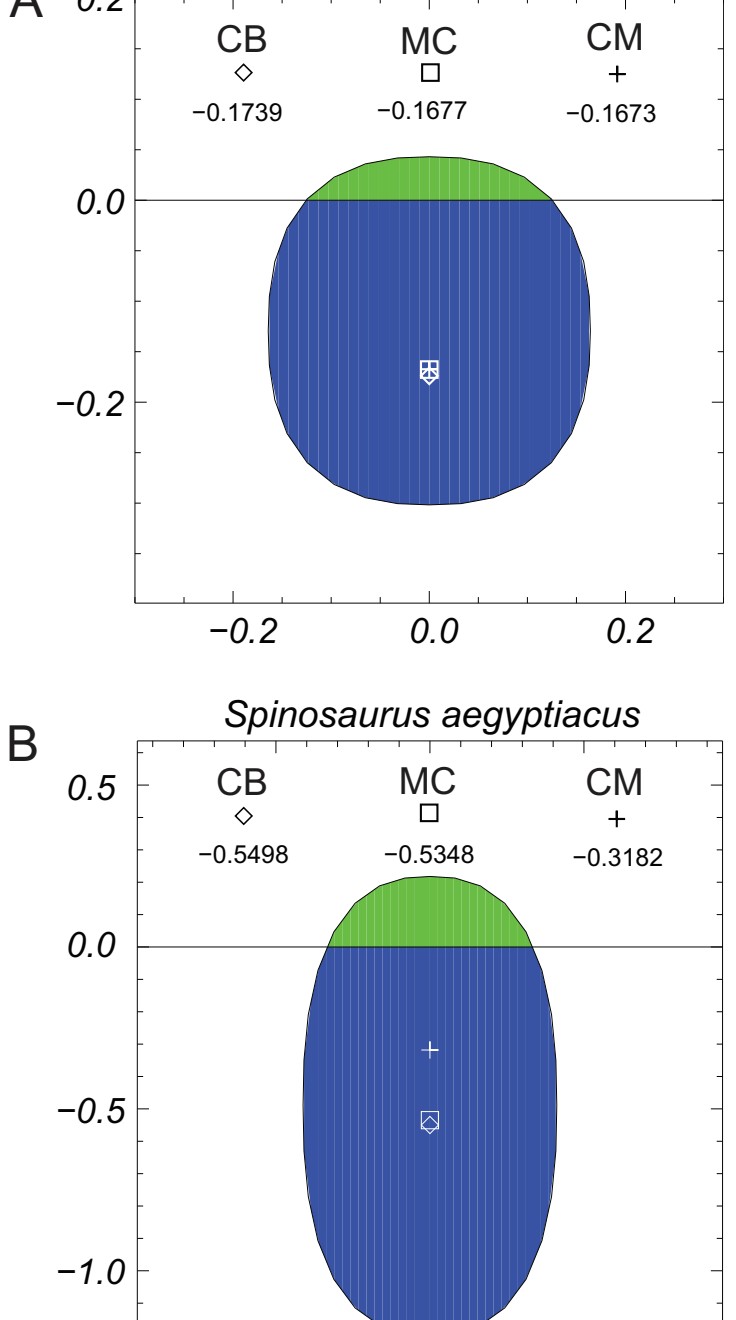

**Figure 7 Graphical views of the metacentric heights (MC '□'), centers of buoyancy (CB '◇'), and centers of mass (CM '+') computed from the three-dimensional models.** (A) *Alligator mississippiensis*; (B) *Spinosaurus aegyptiacus*. A center of mass above the metacentric height indicates an unstable situation, which is clearly the case for the *Spinosaurus*. Stated measurements are relative to the water line and are in meters. See 'Methods and Results' sections for more details. Green indicates the 'dry' area above the waterline, while the blue is the 'wet,' immersed portion.

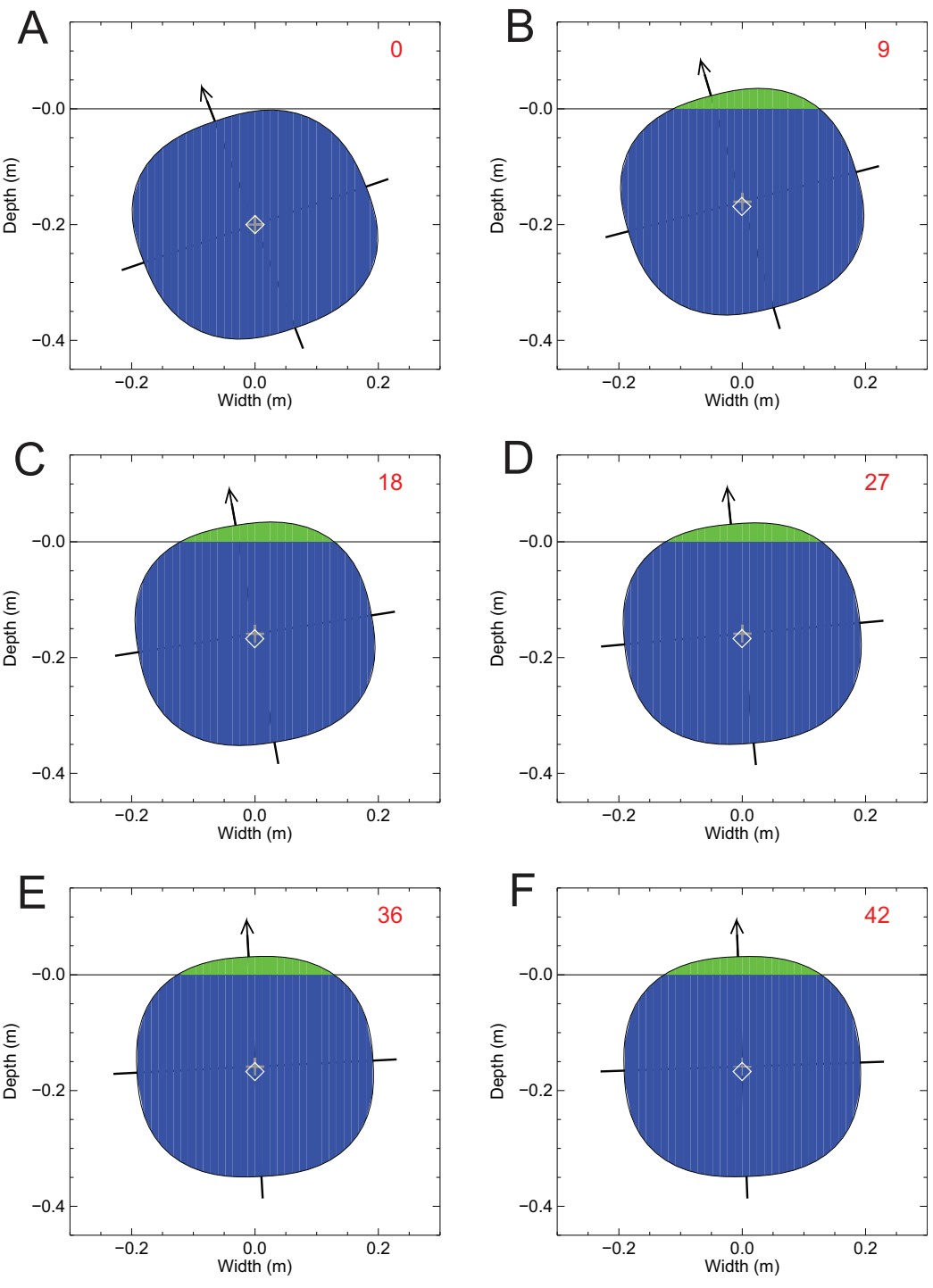

**Figure 8 A test of the lateral stability of the floating *Alligator* model using a disk representing the transverse section of the immersed axial body at the position of the CM from the floating model of Fig. 3B.** The disk was given a 20 sideways tip, but over the course of 42 simulation cycles it slowly returned to an upright orientation by passive self-righting. Symbols and colors as per Fig. 7.

the longitudinal position of the whole body CM. Although not shown, when this disk was placed in water without any lateral tipping, it came to stable equilibrium with 95.25% of the disk immersed and remained upright. The mass density of the disk is 952 gm/l. For the lateral stability test the model disk was tipped sideways by 20° (Fig. 8, frame no. 0). This resulted in a small, although not visible, horizontal separation between the CM (grey '+') and the CB (white '◇'). The shape of the whole cross-section and its immersed portion, and the relative positions of the CM and CB, resulted in the disk returning to equilibrium with the original topside uppermost (Fig. 8, frame no. 42). The vertical positions of the CB and mass, relative to the water surface in this final state were −0.167 and −0.159 m, respectively. The lengthy number of cycles needed to return to equilibrium, 42 (also the answer to 'life, the universe and everything' (Adams, 1982)), is interpreted to be the result of the CM and CB being almost coincident and the moment arm of the restoring buoyant forces being very small, and this was also predicted with the previous computation of the MC (Fig. 7). The final degree of immersion was the same 95.25% as before. This capacity for stability and self-righting when floating at the surface is what could be expected for a semi-aquatic animal that habitually spent extended periods at the water surface. Confirmation of this dynamic stability was observed directly in a floating, and occasionally gently paddling and rolling pair of caimans (*Caiman crocodylus*) that remained upright at the Vancouver Aquarium (Graham Amazon Gallery), Stanley Park, Vancouver, British Columbia (D. Henderson, 2018, personal observation).

When not tipped sideways, the disk representing the *Spinosaurus* cross-section remained upright, with 82.8% immersion. The mass density of the disk is 833 gm/l and ideally the disk should have come to equilibrium with 83.3% immersion. The modelled value of 82.8% is only off by 0.6% of the expected value, and this discrepancy is interpreted to arise from modelling process and the asymptotic nature of how the disk is brought to equilibrium. Figure 9 confirms the instability predicted from the relative positions of the MC and the CM (Fig. 7), and shows what happened when the *Spinosaurus* disk was tipped sideways by 20°—the disk quickly rolled over onto its side, with the final, equilibrium vertical positions of its CB and CM being −0.301 and −0.239 m, respectively. Figure 9 also demonstrates that for assessing lateral stability, the two-dimensional approximation is a valid one in the present situation, and highlights the dominance of the axial body in determining the overall lateral stability. This test shows that the body of a floating *Spinosaurus* would have been liable to tip when nudged, and suggests that *Spinosaurus* must have had to apply constant limb action to maintain an upright posture when in water when subject to any disturbances at the surface. This does not appear to be an attribute of an animal well-adapted for a semi-aquatic life.

Figure 10 shows the fleshed-out form of the model hindlimb of the *Allosaurus* model from Fig. 1 along with its larger limb bones. The volume of the hindlimb mesh was found to be 0.1152 m³, and with the assigned density of 1,050 kg/m³, it has a mass of 121 kg. The bones have a combined volume of 0.01052 m³, and subtracting this from the total

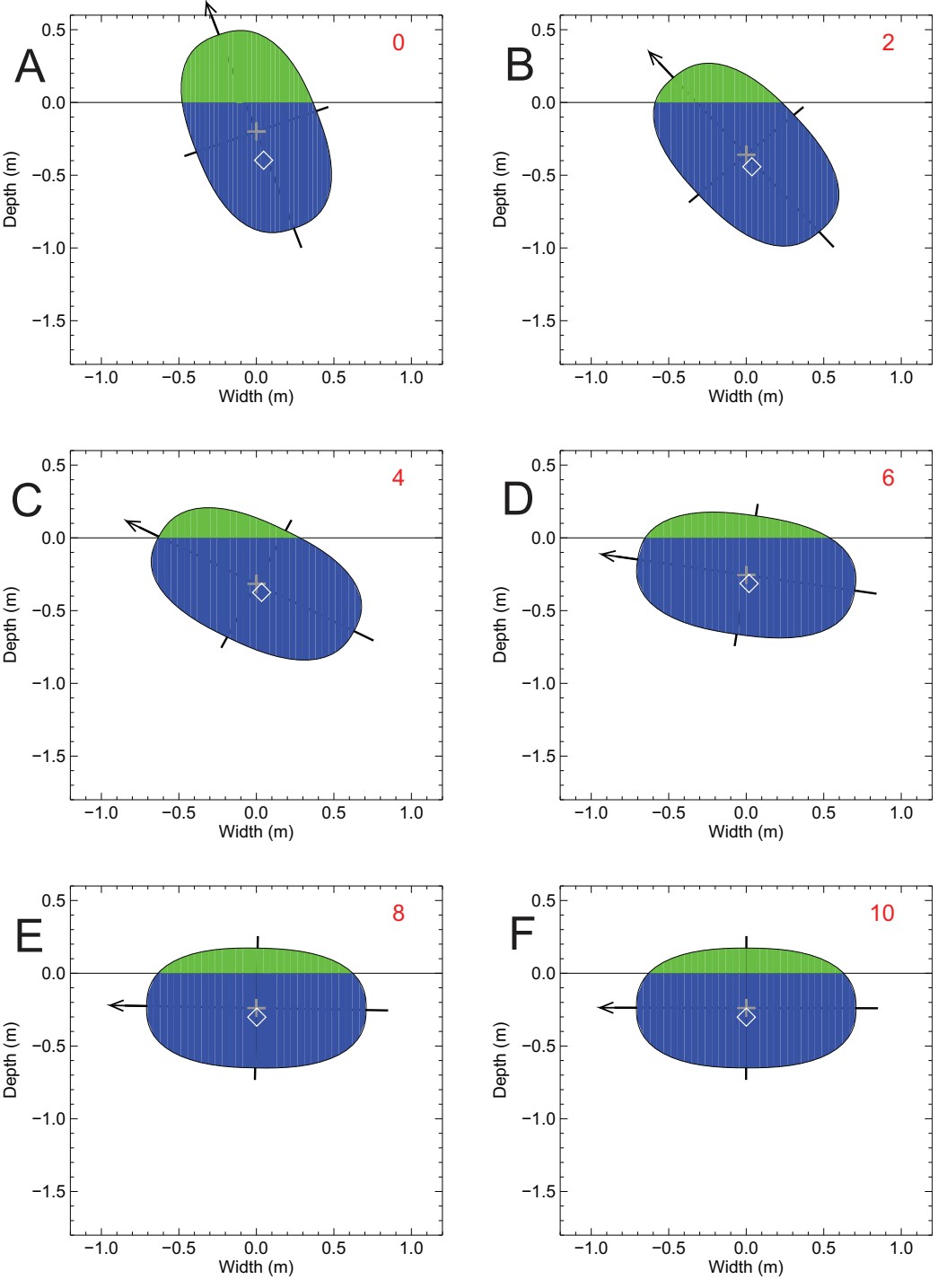

**Figure 9 A test of lateral stability of the floating *Spinosaurus* model using a disk representing the cross-sectional area of the axial body at the position of the CM from the floating model of Fig. 5A.** The disk was given a 20 sideways tip, but over the course of 10 simulation cycles it quickly rolled onto its side to a new position of stable equilibrium. Symbols and colors as per Fig. 7.

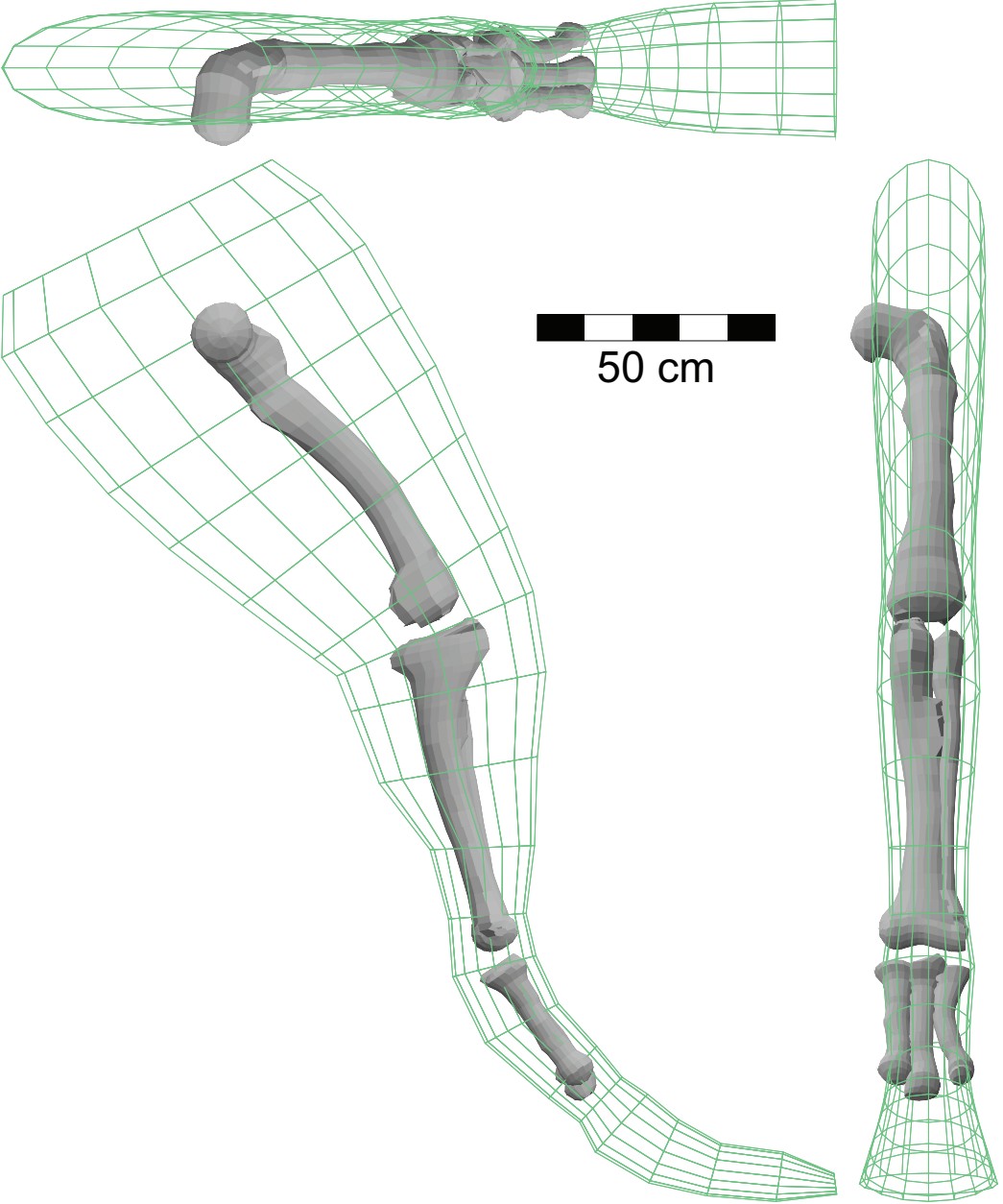

**Figure 10 Isometric views of hindlimb model of *A. fragilis* using the right limb from Fig. 1C and three-dimensional models of the large limb bones based on illustrations in *Madsen (1976).*** The volumes of these shapes, combined with the appropriate densities, were used to investigate the effects of higher than normal bone densities on the mass and density of the host animal. See 'Results.'

volume leaves a flesh (non-bone) volume of 0.1047 m$^3$. The mass of the leg can be expressed as:

$$\text{leg\_mass} = \text{flesh\_volume} * \text{flesh\_density} + \text{bone\_volume} * \text{bone\_density} \qquad (2)$$

Given that the total leg mass and the flesh and bone volumes are known, and assuming that the flesh density is 1,000 gm/l, one can solve Eq. (2) for the bone density.

This gives a bone density for the *Allosaurus* leg model of 1,547 gm/l, approximately 50% more than that of water. Assuming that compact bone has a density of approximately 2,000 gm/l, the reduced bone density derived from Eq. (2) is consistent with an open medullary cavity in the bones. The mass of the bones is computed as their volume multiplied by their density, and comes to 16.28 kg. The bones of the single hindlimb represent 1.69% of the total body mass estimated for the *Allosaurus* model of 963 kg (Table 2). With the availability of the bone volumes, the effects of increasing the density of the bones to increase their mass can be analysed. Assuming that the bones are solid, as observed with *Spinosaurus* (*Ibrahim et al., 2014*), and with a density of 2,000 gm/l, one gets a heavier bone mass of 21.04 kg which now represents 2.248% of total body mass, an increase of just over half of 1% of total mass. The leg of the new restoration of *Spinosaurus* is estimated to have a mass of 295 kg, more than twice that of the *Allosaurus* model. However, the body mass of the *Spinosaurus* at 6,379 kg is almost seven times as great as that of *Allosaurus*, and the hind limb represents just 4.54% of total body mass. Assuming the same bone to flesh proportions in the hindlimbs of *Spinosaurus* and *Allosaurus*, any increase in the mass of the relatively smaller hindlimbs of *Spinosaurus* via solid bones will be an even smaller fraction of total body mass than that estimated for *Allosaurus*. Given the inherent uncertainties of the various densities of the various body regions, and their true volumes, exceptional evidence would be needed to demonstrate that the increase in body mass by a few percent by having denser limb bones would significantly affect the ability of a *Spinosaurus* to immerse itself.

## DISCUSSION

*Ibrahim et al. (2014)* list details of *S. aegyptiacus* and its ancient environment that plausibly suggest this dinosaur was specialized for a semi-aquatic mode of life. These details include: highly unusual adaptations such as a higher bone compactness than seen in alligators; peculiar morphology of the pes; extremely retracted position of nares; very few remains of plant-eating dinosaurs in the Kem Kem beds and other equivalent sequences in North Africa; and the presence of abundant giant fishes presenting seemingly optimal conditions for large, fish-eating tetrapods and fish-based food webs. However, while no amount of evidence can prove the validity of a hypothesis, it only takes one contradictory observation to potentially falsify it. The three problems with the hypothesis of a semi-aquatic *Spinosaurus* identified in the current work would appear to seriously weaken the hypothesis of Ibrahim et al., and these are discussed below.

Contrary to the claim by *Ibrahim et al. (2014)* that the CM of *Spinosaurus* was centrally located in the trunk region, this study finds the CM much closer to the hips than previously estimated. In fact, it is less than the relative CM distance determined for the *Tyrannosaurus* model (Table 2—horizontal relative CM position). This is interpreted to be a consequence of the new restoration of *Spinosaurus* and the associated muscle mass of its substantially longer tail when compared to that of *Tyrannosaurus*. Having a CM closer to its hips indicates that *Spinosaurus* would still be competent as a terrestrial biped since the CM would be above and/or between the supporting feet while walking (*Henderson & Nicholls, 2015*). A validation of the present method for determination of the

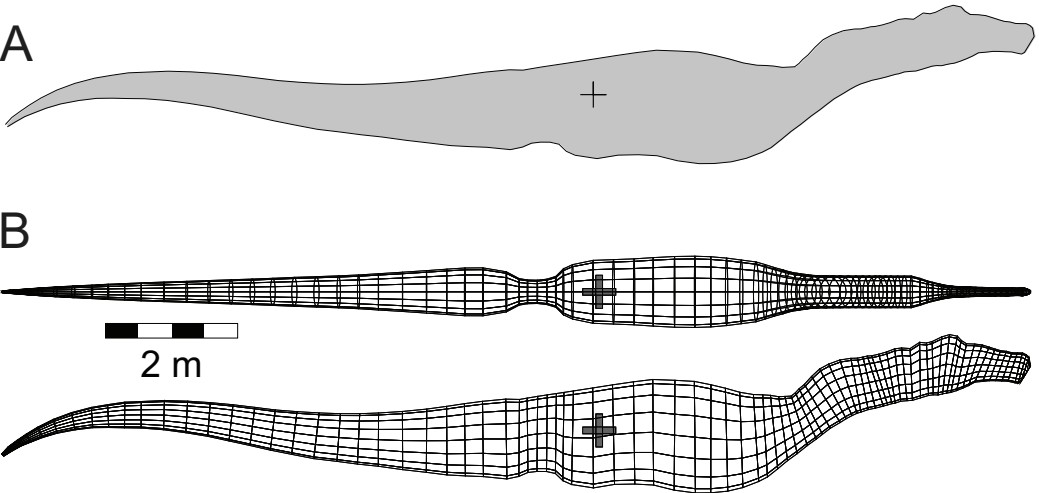

**Figure 11 Centres of mass determinations for the axial body of *Spinosaurus* using two different methods.** (A) Two-dimensional silhouette with constant areal density; (B) three-dimensional mesh without lung cavity or air sacs. In neither case does the CM reside at the midpoint of the trunk region as claimed by *Ibrahim et al. (2014)*. See 'Discussion' section.

CM in theropods comes from an estimate of the CMs of a standing pigeon and ostrich (*Henderson, 2010*). With densities appropriate for birds assigned to the heads, necks, trunks, and limbs of models of the latter two animals, their CMs were found to lie directly above and between the feet enabling the animals to stand in a stable fashion (*Henderson, 2010*, fig. 1), as can be observed in the living forms.

In an attempt to replicate the more anteriorly located CM for *Spinosaurus* reported by *Ibrahim et al. (2014)*, two alternate versions of determination the CM were tried. The first attempt involved determining the centroid of the two-dimensional lateral profile of the axial body. This 2D centroid is located towards the rear of trunk region, and slightly posterior to the ventral bulge associated with the pubis (Fig. 11A). A second attempt used just the axial body of the three-dimensional model and assumed a uniform density, no pneumatic cavities, and no lung cavity. The resulting 3D CM was again located towards the rear of the trunk region, but just ahead of the ventral bulge associated with the distal end of the pubis (Fig. 11B). None of the three computed values for the CM for *Spinosaurus* in the present study can match that reported by *Ibrahim et al. (2014)*.

*Spinosaurus* is certainly able to float and breathe with the head above water (Fig. 5A). However, there is nothing special about the state of an immersed *Spinosaurus*. With the modest 15° upwards tilt of its head relative to that of the basic terrestrial form (Fig. 1F), the *Tyrannosaurus* is also able to float and breathe (Fig. 6D). Furthermore, the *Tyrannosaurus* model is 51% heavier and slightly denser than the *Spinosaurus* one (Table 2), yet is able to keep most of the head clear of the water surface. The floating equilibrium states of the four other, lighter models—*Baryonyx* (*Suchomimus*) (Fig. 5B), *Coelophysis* (Fig. 6A), *Struthiomimus* (Fig. 6B), *Allosaurus* (Fig. 6C)—are consistent with the floating states of the two, heavier, longer animals. These results are not unexpected, as

most terrestrial tetrapods can successfully float and swim (see *Henderson & Naish, 2010* for review).

It was found that the alligator model would sink when the lungs were deflated by 40–50% (*Henderson, 2003b*). However, the lower mean densities of the two spinosaurid models, relative to that of the alligator, immediately suggests that they might not be able to sink and become fully immersed. This was tested by deflating the lung of the *Spinosaurus* model by 75%. This had the effect of increasing the mean density of the model from its basic value of 823 gm/l to 885 gm/l. It should be mentioned that the lung deflation process was associated with an elevation of ventral abdominal region of the model body so that the volume decrease of the axial body was reflecting the volume decrease of the lung. With the increased density, the model reached buoyant equilibrium at the lower depth of 48 cm, compared to the 37 cm when the lung was fully inflated. However, the new density is still less than that of water, 1,000 gm/l, indicating that the animal would still be buoyant. Extant semi-aquatic birds and reptiles such as penguins, loons, ducks, cormorants, sea snakes, marine iguanas, crocodilians, and both marine and freshwater turtles ALL have the ability, and the apparent need, to become submersed to enable the pursuit of prey, or in the case of the marine iguana, forage on the sea bed. The same is true of semi-aquatic mammals such as otters, musk rats, waters shrews, beavers, hippos, and polar bears. Not being able to become fully immersed for any of these taxa listed would be a major impediment. The inability of a *Spinosaurus* to sink underwater would severely limit its ability to effectively capture aquatic prey, and conflicts with the suggestion that *Spinosaurus* was specialized for a semi-aquatic life when *Ibrahim et al. (2014)* explicitly state '... in the pursuit of prey underwater' (sentence four, paragraph five).

As a test of how sensitive the buoyant *Spinosaurus* model was to the assumed presence of avian style air sacs and pneumatized bone, an alternate model lacking these features was tried. This model assigned a uniform axial density of 1,000 gm/l from the tip of the tail to the tip of the snout. The limb and sail densities were unchanged, and the same lung was retained. This alternate model can also be thought of as one with a denser skeleton. This model has higher mean density of 918 gm/l and is heavier, 7,160 kg, than the standard one with its density of 833 gm/l and mass of 6,379 kg. Deflating the lungs of this denser model by 75% resulted in an even greater mean body density of 986 gm/l, and deeper depth of immersion for the CM at 0.696 m, but this model was still not able to sink as its density was still less than that of fresh water. If it could be shown that the mass deficit represented by the lungs and air sacs was offset by the increased mass of a denser skeleton that might help the claim of a semi-aquatic *Spinosaurus*.

It should not be forgotten that the restoration of *Spinosaurus* by *Ibrahim et al. (2014)* is based on the composition and scaling of the remains of several animals from different localities, along with missing details supplied from other spinosaurids such as *Baryonyx* (*Suchomimus*), *Irritator,* and *Ichthyovenator* (caption for Fig. S3, *Ibrahim et al., 2014*). In particular, the hind limbs of the new restoration, although from a single individual, were not associated with a complete dorsal axial skeleton. The colour codings of the vertebrae used in the reconstruction (*Ibrahim et al., 2014*, Fig. S3) clearly show that the majority of the vertebrae come from other animals and locations. The only

partially contiguous set of vertebrate are those of the anterior and mid-dorsals and the incomplete sacrum from the original specimen described by *Stromer (1915)*. With the axial body providing the majority of the body mass, any systematic errors in the restoration of body length will affect estimates of total body mass and relative limb/body proportions. The restored hindlimb proportions of *Spinosaurus* do appear to be rather small when compared to the rest of the body, and when compared with the hind limb-body proportions seen in other theropods. Figure 12 shows a plot of relative masses of single hind limbs, expressed as a percentage of total body mass for the six animals of the present study. For the computation of the mean and standard deviations shown in Fig. 12, the values for *Spinosaurus* were not included. The relative hindlimb mass of the restored *Spinosaurus*, 4.88%, is less than half the mean relative mass computed for the other five of 12.6% (stan.dev. = 1.87%). It might be argued that the qualitative reconstructions of the forms of the hindlimbs of the models might be highly subjective, and subject to bias. However, some qualitative aspects of the plot argue for its general plausibility. *Struthiomimus*, interpreted to be highly cursorial (*Russell, 1972*), and assumed to have extensive hindlimb musculature for running, has the highest relative leg mass with it plotting more than one standard deviation above the mean (the dashed line of Fig. 12). The heaviest animal of the present study, *Tyrannosaurus*, has the second highest relative limb mass, while lightest animal, *Coelophysis*, has a relative leg mass less than the mean value.

Modern, semi-aquatic crocodilians have relatively smaller hind *and* forelimbs when compared to their more terrestrial ancestors such as *Sebecus* (*Pol et al., 2012*) and *Terrestrisuchus* (*Crush, 1984*). This reduction in limb size is interpreted as an adaption to reduce drag while swimming, and reflects the dominance of axial musculature for aquatic propulsion (*Grigg & Kirshner, 2015*). If the reduced hindlimbs of the new restoration of *Spinosaurus* are an indication of a more aquatic mode of life (*Ibrahim et al., 2014*), one would expect that the forelimbs would also be reduced, similar to what is seen in the crocodilians. However, the forelimbs as restored for Spinosaurus are large enough to reach the ground. Complete forelimbs were not found in association with the hindlimbs or the axial body, and the colour codings in the Supplementary Information Fig. S3 of *Ibrahim et al. (2014)* clearly demonstrates the disparate origins of the forelimb elements in the new restoration. The only two minor exceptions to the mixed origins of the forelimb elements comes from a manual phalanx 2 and an incomplete base of phalanx 3 from digit II that were found with the new specimen. If isometric scaling based on the dimensions of these two elements was used to set the sizes of the other bones, then it needs to be demonstrated that the assumed scaling relationship is valid. The exceptionally large size of *Spinosaurus* compared to other theropods indicates that non-linear, non-isometric changes in bone sizes and their relative proportions in the forelimbs are a distinct possibility, and this undermines confidence in the new restorations.

Despite the above problems with having *Spinosaurus* as an animal that spent substantial amounts of time immersed in water, it is still reasonable to interpret the animal as having some connection with aquatic environments. *Charig & Milner (1997)* noted the gharial-like aspects of the skull and dentition of another well-known spinosaurid,

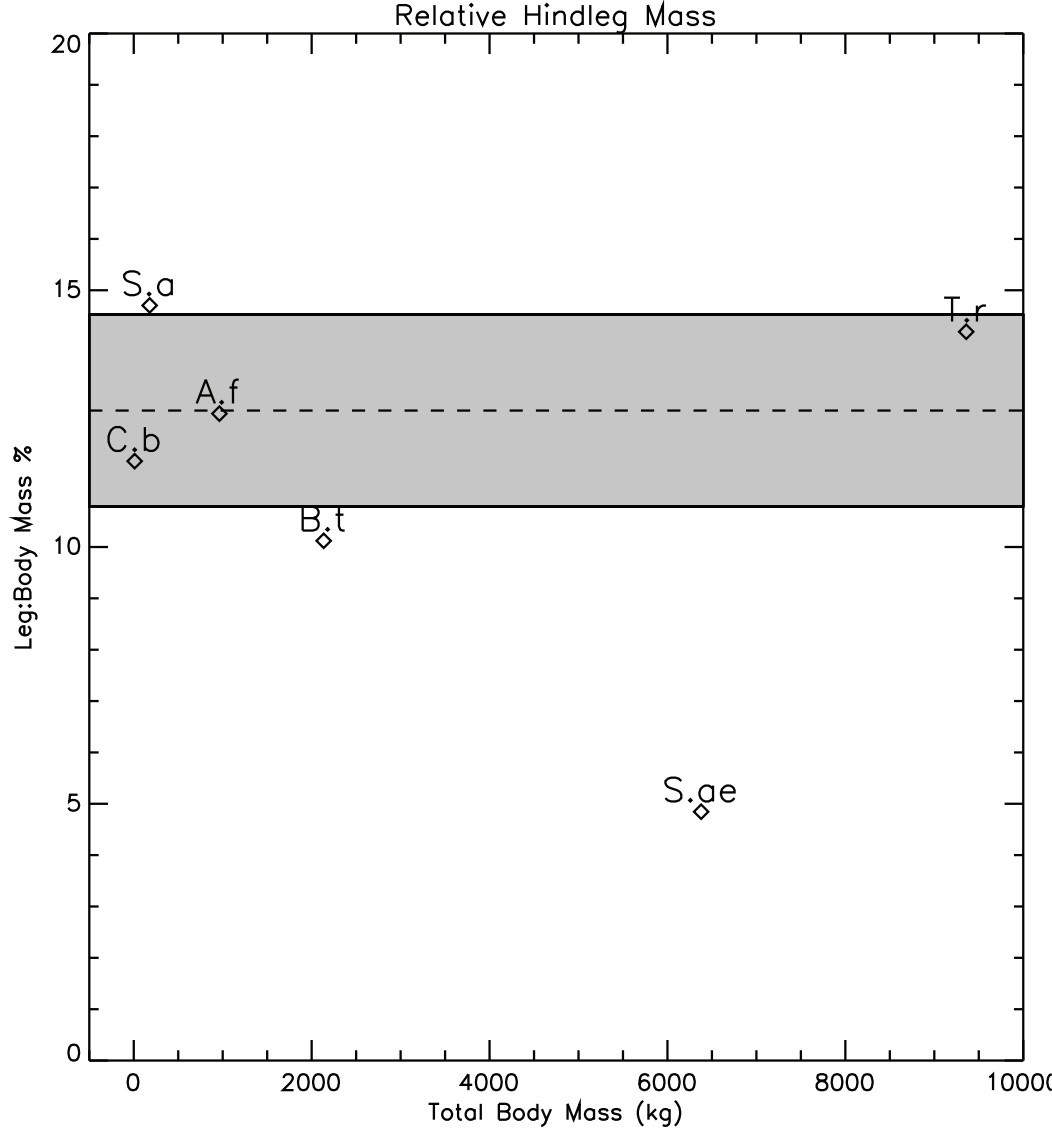

**Figure 12** **Relative mass fractions of the hindlimbs of the theropods in the present study highlighting the small size of the restored *Spinosaurus* hindlimbs.** Dashed line represents the mean value of 12.6%. Grey band spans plus and minus one standard deviation about the mean. The *Spinosaurus* limb mass was not used in the calculation of the mean and standard deviation. A.f, *Allosaurus fragilis*; B.t, *Baryonyx (Suchomimus) tenerensis*; C.b, *Coelophysis bauri*; S.a, *Struthiomimus altus*; S.ae, *Spinosaurus aegyptiacus*; T.r, *Tyrannosaurus rex*.

*B. walkeri*, and proposed that *Baryonyx* was wading in the shallows snatching fish with its specialized jaws. The very robust arms and manual claws of *Baryonyx* were also suggested as another way for the animal to procure aquatic prey without having to become fully immersed—similar to modern grizzly bears (*Charig & Milner, 1997*). *Amiot et al. (2010)* used stable isotope geochemistry analysis of oxygen in the teeth of spinosaurids to show that they must have spent significant time in water and must have included some aquatically derived prey as part of a more generalist diet. *Ibrahim et al. (2014)* made a series of observations of the skull and teeth of *Spinosaurus* that suggested it was well adapted to

sense, pursue, and capture aquatic prey. However, given the findings of the present study, the more conservative, and more terrestrially linked, *Baryonyx* model of Charig and Milner would also seem to be the one for the interpretation of the mode of life of *S. aegyptiacus*.

## CONCLUSION

The combination of a CM close to the hips that still enabled effective terrestrial locomotion, an inability to become negatively buoyant, and a body (when immersed) with a tendency to roll onto its side unless constantly resisted by limb use, suggests that *Spinosaurus* was not highly specialized for a semi-aquatic mode of life. Furthermore, the floating characteristics of the *Spinosaurus* model were similar to those of models of other predatory dinosaurs, indicating that there was nothing special about the buoyant characteristics of this animal, and that other theropods could have successfully taken to water to the same degree as well. Terrestrial activity would still have been part of its normal life of *Spinosaurus*, similar to the interpretations given for other large predatory dinosaurs. Lastly, the new reconstruction of *Spinosaurus* is based on a composition of remains from multiple individuals of varying sizes and proportions that come from different locations, and were scaled to match the presumed proportions of a single individual. This does not seem like a good platform for building hypotheses about what this animal was like as a once living organism.

## ACKNOWLEDGEMENTS

I thank Jim Gardner (Royal Tyrrell Museum of Palaeontology) for reading an earlier draft of the text. I am most grateful to the reviewers, in particular C. Palmer, V. Allen, and the I. M. Anonymous twins whose questions, constructive comments and suggestions made me think more carefully about what I wanted to say and how to say it.

### Funding

The author received no funding for this work.

### Competing Interests

The author declares that he has no competing interests.

### Author Contributions

- Donald M. Henderson conceived and designed the experiments, performed the experiments, analysed the data, contributed reagents/materials/analysis tools, prepared figures and/or tables, authored or reviewed drafts of the paper, approved the final draft.

### Data Availability

The raw data are provided in the Supplemental Files.

## Supplemental Information

Supplemental information for this article can be found online at http://dx.doi.org/10.7717/peerj.5409#supplemental-information.

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
