# Peer review of "A buoyancy, balance and stability challenge to the hypothesis of a semi-aquatic Spinosaurus Stromer, 1915 (Dinosauria: Theropoda)"

_PeerJ, doi:10.7717/peerj.5409_

## Round 0.1 · original submission · Minor Revisions

Dear author,

I have accepted the decision of 'minor revisions' from reviewers one and three, however please note reviewer two did recommend ‘major revisions’ and has made a series of suggestions. As reviewers one and three discussed different aspects of the sensitivity analyses and furthering them, this would be an ideal place to explore the differences in bone density mentioned by reviewer two.

Furthermore, I have some additional comments that the authors should address prior to resubmission (in addition to those made by the reviewers):

1. Authority and date should be provided for each species-level taxon at first mention (including in the title, abstract and introduction). Please ensure that the nominal authority is also included in the reference list.
2. In the title, can you include a higher taxon name.

Once again, thank you for submitting your manuscript to PeerJ and I look forward to receiving your revision.

·

Basic reporting

Clear and unambiguous, professional English used throughout.

Literature references, sufficient field background/context provided.
Appears so, but note that I am not a Spinosaurus expert.

Professional article structure, figs, tables. Raw data shared.
Structure is good. (Small point: LL 219 to 233 might be more logically placed in the methods section.)
Figures good.
Data sharing is adequate - but see comments below under Methods.

Experimental design

Original primary research within Aims and Scope of the journal.
Yes

Research question well defined, relevant & meaningful. It is stated how research fills an identified knowledge gap.
The study addresses the problem of reconstructing the Spinosaurus, which has previously been described as a fully aquatic predator. By using a well proven modelling approach, the author throws considerable doubt upon this view.

Rigorous investigation performed to a high technical & ethical standard.
The modelling approach has been used by the author in other published studies and appears to be robust and reliable. As I do not have access to the modelling software, I cannot confirm this.

I have two points related to the modelling, which are really my only significant comments on the paper:

1) It is incorrect to say that “The second criterion was that the centre of mass (CM) and the centre of buoyancy (CB) be either almost coincident, or that the CM lie below the CB, in order that the model not capsize.” Ship designers have known since the 1730s (see Ships and Science by Larrie D Ferreiro, chapter 4) that the stability of a floating body is determined by the height of the metacentre above the centre of gravity, where the metacentre is the point where the line of action of the buoyancy force crosses the centreline as the body is inclined laterally (or indeed longitudinally). The CM can be above the CB and a ship will still be stable (indeed, this is the case for the great majority of vessels.) This is described in detail in Comstock 1967 (Chapter 2) but can be found in any textbook on ship or boat design.

It would be useful if the author were to calculate the lateral stability of the models and discuss the results. I imagine it would be a simple matter to adapt the software to calculate the second moment of area of the water-plane and thus the location of the metacentre.

2) Sensitivity. The author describes his assumptions in considerable detail and provides good justification for them. However, they can only ever be informed estimates, so it would be helpful if some indication of the sensitivity of his results to his assumptions were given in the text.

Methods described with sufficient detail & information to replicate.
Replication would require access to the PV-Wave program, which I do not have. However, all the indications are that the author has provided sufficient data for replication to be possible for those with access to this program.


References

Ferreiro, L. D. 2007. Ships and Science: the birth of naval architecture in the scientific revolution, 1600-1800. MIT Press, London.

Comstock, J. P. (ed) 1967. Principles of Naval Architecture. SNAME, New York.

Validity of the findings

Data is robust, statistically sound, & controlled.
See comments above re sensitivity and need to discuss lateral stability.

Conclusion are well stated, linked to original research question & limited to supporting results.
Yes

Additional comments

From a technical, biomechanical perspective this very well argued piece of work. I am not very knowledgeable about the Spinosaurus debate, so cannot useful comment on that aspect of the work.


There are a few small typos etc:

The caption for Figure 5 contains an error. Presumably 9360 kg, not t!
Figure 5 Floating theropods with masses ranging from 10.3 kg to 9360 t 

There are a number of random extra spaces (and missing spaces) in figure captions, also in Abstract. This may be an artefact of the conversion to PDF?Some species names not italicised. Most of the relevant text is copied below.

large predatory dinosaur Spinosaurusaegyptiacus proposed

However, similarly detailed models of another spinosaurid Baryonyx ( Suchomimus ) tenerensis, along with models of the more distantly related Tyrannosaurus rex , Allosaurus fragilis , Struthiomimus altus and Coelophysis bauri were

(A) Coelophysis bauri ; (B) Struthiomimus altus ; (C) Allosaurus fragilis ; (D) Baryonyx ( Suchomimus ) tenerensis ; (E) Spinosaurus aegyptiacus ; (F) Tyrannosaurus rex . Animals
Detailedview of the Spinosaurus “sail” andits associated neural spines
Three-dimensional alligator (Alligator mississippiensis ) model
(A) Spinosaurus aegyptiacus ; (B) Baryonyx ( Suchomimus ) tenerensis . Determination
(A) Coelophysis bauri ; (B) Struthiomimus altus ; (C) Allosaurus fragilis ; (D) Tyrannosaurus rex . See
AlternateSpinosaurus axial
A.f – Allosaurus fragilis , B.t – Baryonyx ( Suchomimus ) tenerensis , C.b – Coelophysis bauri , S.a – Struthiomimus altus , S.ae – Spinosaurus aegyptiacus , T.r – Tyrannosaurus rex .
Hatcher JG. 1901. Diplodocus Marsh, its osteology, taxonomy, and probable habits, with a restoration of the skeleton. Memoirs of the Carnegie Museum 1:1-64.

Reviewer 2 ·

Basic reporting

Important references are missing, and the background provided is incomplete. Major problems are listed below.

- Important and highly relevant research is not cited and/or discussed - a major problem with this manuscript is that the author does not (or not sufficiently) address several very important issues, including 1) highly unusual adaptations in Spinosaurus used as evidence for a more aquatic lifestyle (e.g., bone histology results (higher bone compactness than Alligator); peculiar morphology of the pes; extremely retracted position of nares – see for example Ibrahim et al. 2014), 2) oxygen isotope analyses strongly suggesting a semiaquatic lifestyle (see for example Amiot et al. 2010), and 3) paleoecological aspects (e.g., presence of abundant giant fishes; very few remains of plant-eating dinosaurs in the Kem Kem and other equivalent sequences in North Africa; seemingly optimal conditions for large, fish-eating tetrapods and fish-based food webs – see for example Läng et al. 2013, Benyoucef et al. 2015).

- The content and narrative of the MS suggests that this is a critique of a “diving Spinosaurus” proposed by Ibrahim et al. 2014. A reading of Ibrahim et al. (2014), however, shows that these authors merely argue for semiaquatic habits, diving is not mentioned. They state, regarding the neck of Spinosaurus: “The horizontal cervicodorsal hinge created by these broad centra would facilitate dorsoventral excursion of the distal two thirds of the neck and skull in the pursuit of prey underwater”. This would be in agreement with the “floating Spinosaurus” presented here, so it is unclear why this MS is presented as a “critique” of the 2014 paper. Maybe the author is critiquing other papers relating to this dinosaur? Biophysicist Gimsa et al (2016), for example, specifically discuss the sail use and diving abilities of Spinosaurus.

- Page 9, line 76: Again, semiaquatic habits for spinosaurids have been proposed well before the “radical” 2014 paper by Ibrahim et al. For example, Amiot et al. (2010) have used detailed oxygen isotope analyses to come to the same conclusion. These other papers and lines of evidence definitely have to be thoroughly addressed here.

- Page 17, line 262: This claim appears to be inaccurate. According to Ibrahim et al (2014) the neotype skeleton includes at least some associated dorsal elements (vertebrae, spines).

Page 19, line 287: Again, this seems to contradict the published record. Ibrahim et al (2014) list at least one very large forelimb element (manual phalanx) associated with the axial and appendicular skeleton.

A few comments on wording, references etc.:

- Page 7, line 52: “Based on a skeletal reconstruction derived from several, isolated, partial specimens of the Late Cretaceous dinosaur Spinosaurus aegyptiacus,”. This statement lacks precision. According to Ibrahim et al. (2014) the reconstruction is based on one partial, associated skeleton + isolated specimens from other localities. The current wording is confusing.

- Page 9, line 91: Make sure to order references chronologically here.

- Page 18, line 278: This should read “the lightest animal”

- Page 19, line 307: Reword this sentence.

Experimental design

The author might want to provide more details on his reconstruction of external and internal soft tissues for Spinosaurus which appears to differ from other theropods in many features, including the (presumably) very small fleshy nares for an animal this size. Why, for example, was the size and distribution of air sacs assumed to be the same for all the theropods?

The theropod models used here all seem to be based on the same parameters (in density for example). This approach does not seem to take into account that Spinosaurus had denser bone than other theropods (see Ibrahim et al. 2014). Clearly, the models need to take into account such factors. Again, the author might want to examine the remains of Spinosaurus in person to ascertain bone density in different parts of the skeleton.
See for example this paragraph from Coughlin and Fish (2009), Journal of Mammalogy: “Increasing body density by increased deposition of compact bone in the appendicular skeleton has been cited as a means that semiaquatic mammals can use to increase their specific density to overcome buoyancy (Fish and Stein 1991; Wall 1983). Osteosclerosis is an increase in bone density by the replacement of cancellous bone with compact bone or by increasing cortical bone thickness at the expense of the medullary cavity (Domning and de Buffrénil 1991; Thewissen et al. 2007; Wall 1983). Osteosclerosis is a common adaptation in semiaquatic and aquatic mammals for buoyancy control (Domning and de Buffrénil 1991; Fish and Stein 1991; Gray et al. 2007; Wall 1983). Additional bone density is achieved by tightly packing cancellous bone into the medullary cavity, as occurs in H. amphibius (Wall 1983).”

It would be helpful and important to compare Spinosaurus to a hippo, a semiaquatic animal with high-density bone that is considered an excellent diver (Coughlin and Fish 2009). How does the buoyancy of a hippo compare?

Other problems with the models used:

- Page 15, line 218: The proportions (and, possibly, density) of Spinosaurus appear to be very different from those of pigeons and ostriches. The author might want to explain how those differences have been taken into account.

- Table 2: The body length estimate for Spinosaurus (16 m) provided in the MS seems to be longer than the ~15 m estimate by Ibrahim et al (2014).

A few more issues:

- Page 8, line 56: Contrary to one or two usages of the term in the literature, the Kem Kem sequence has never been defined as a formation. The author might want to use the term “Kem Kem beds” instead. See also Cavin et al. (2010) for an alternative interpretation of the sequence.

- Page 8, line 57: “This interpretation of an extinct theropod as being semi-aquatic was entirely novel, and generated much media attention (eg. Tarlach, 2014; Coghlan, 2014).” Media attention is probably not a relevant factor in this technical publication and should not be mentioned in the context of scientific issues.

Validity of the findings

In addition to methodological issues and missing background infromation, there are also some other problems with the conclusions:

- That theropods could float (and swim) is expected and consistent with previous work and ichnofossils. Unfortunately the author does not shed light on the reasons why Spinosaurus shows so many anatomical features typically associated with increased semiaquatic, or aquatic, lifestyles in other vertebrates. More importantly, if Spinosaurus is “like Baryonyx”, why is its skull and postcranial anatomy so different and – according to many researchers – consistent with increasingly aquatic habits? These major issues, which are at the heart of recent reconstructions, are not sufficiently addressed.

- Page 20, final part of conclusion: “Lastly, the new reconstruction of Spinosaurus is based on a composition of remains from multiple individuals of varying sizes and proportions that come from different locations, and were scaled to match the presumed proportions of a single individual. This does not seem like a good platform for building hypotheses about what this animal was like as a once living organism.”

But is this not what paleontologists have to work with much of the time? Indeed, the author (Henderson, 2010) has used the same approach when reconstructing the weight and flight abilities of the giant pterosaur Quetzalcoatlus, a taxon known from (often very) incomplete remains representing multiple individuals of varying sizes, from different locations in Texas’ Big Bend National Park (Javelina Formation), and frequently scaled to match the presumed proportions of a single individual (see Henderson 2010)…

Additional comments

The author attempts to shed light on the buoyancy and balance of a bizarre predatory dinosaur, Spinosaurus aegyptiacus. This is an interesting modeling approach to a potentially unique animal and includes interesting aspects, such as a reassessment of the center of mass. Unfortunately the manuscript, in its current form, is lacking in a number of very important areas and will require substantial revision before it can be considered for publication. For details on how best to improve different aspects of the manuscripts see comments and references in the relevant review sections.

It is also not clear from the manuscript whether the author examined the Spinosaurus material in person. Presumably, this would be very important to ensure that the model is as accurate as possible (e.g., first-hand study of spines and long bones, establishing bone density values for different parts of the skeleton, etc).

·

Basic reporting

Clear English is used throughout, bar the following minor issues:
- Line 41: “e.g. examples” I think reads better than “examples, etc.”
- Line 149: Is the comma between ‘large’ and ‘reptile’ a typo?
- Line 211: ‘Contrary to Ibrahim et al. (2014) that the centre’, it seems like a word may be missing here (something like ‘Contrary to Ibrahim et al.’s (2014) statement’?)

Sufficient literature references and context are provided.

The article structure is logical and reads well. Raw data is shared, but inclusion of the polygon meshes used to reconstruct body mass properties would be nice for repeatability / investigation using reconstruction techniques other than the authors outline-based ‘mathematical slicing’.

The article is appropriately self-contained, given that the method has been well-explained elsewhere in previous papers.

Experimental design

This is original primary research and by my judgement it fits within the aims and scopes of the journal.

The research question is well-defined, relevant and meaningful to the field of dinosaur behaviour reconstruction, both in terms of its results and its methodology.

The investigation is sufficiently rigorous and is at the upper end of the technical methods for the field. Ethics concerns are not applicable.

The methods are well described and replicable, albeit only with reference to the authors earlier works. I do not consider the latter a problem.

Validity of the findings

This is essentially ‘meaningful replication’, in that it is an attempt to replicate the centre of mass estimated of Ibrahim et al.’s 2014 study, but using an explicitly explained (albeit in earlier work) and well-validated method, rather than the unexplained method of Ibrahim et al. As Ibrahim et al. made what could be considered extraordinary claims (a quadrupedal, habitually aquatic non-avian theropod), but do not state their method fully, an attempt to replicate their findings is fully justified. In addition, the centre-of-buoyancy calculations are fascinating in themselves, and definitely add meaningful data to the field.

The robusticity of the data is I think sufficient. The author goes to some lengths to vary his assumptions in order to disprove his main findings. However, the validation of his methodology would be (to my mind) more convincing if an aquatic theropod (a duck maybe?) were included along with the alligator model. This also pertains to some of the statements made in defence of a non-aquatic Spinosaurus (see general comments below). The descriptive statistics used to summarise the data are simple but entirely appropriate for the sample.

The conclusions of the study are well-considered and well-stated (but see general comments below), and linked well to the primary research question(s) of the paper.

Speculation is minimal (although I have some suggestions in that regard, see general comments) and identified where present.

Additional comments

I congratulate the author on a well-conceived and well-executed study that addresses a pertinent question in dinosaur behavioural research, reported with a clear, concise and very readable manuscript. I do however have some suggestions that I think may improve the study. Firstly, there is no mention made of diving birds – how does their density / pneumatisation compare to that reconstructed here for a typical non-avian theropod? How does this affect their buoyancy both with/without the lungs collapsed? I would be interested to see how a model based on a living aquatic, predatory theropod would compare to the model of Spinosaurus in terms of unsinkability. I would be interested to see how much the density of the Spinosaurus model would have to be increased by in order for it to sink. If given an alligator-style, unpneumatised chest, how would this affect your results? Similarly, while the authors sensitivity analysis is satisfying, I would be interested to hear the author speculate as to what model parameters would be necessary to match the results of Ibrahim et al. (2014).

Also, several of the criteria used as evidence against aquatic behaviour (tightly-fitted acetabulum, stiff abdomen) also exist in swimming birds. The argument against swimming using limb-reduction and streamlining is, also, based on the style of axial propulsion seen in modern crocodilians, and so it seems somewhat tautological so use this for argument an animal like Spinosaurus that is already stated unlikely to be capable of significant lateral body-and-tail flexion. What about paddling? I think a sentence or two addressing this would benefit the article.

Finally, I think Figure 7 may be unnecessary. The summary statistics it represents can be (and are) adequately communicated in the text.

I enjoyed reading this article and look forward to seeing it published. If the author wishes to discuss any of the points raised in this review they are welcome to email me on [email protected].

Dr. Vivian Allen

---

## Round 0.2 · Minor Revisions

Dear author,

I have accepted the decision of 'minor revisions' from two of the three reviewers. Please take on board the comments from reviewers one and two. Both have made comments that will need to be addressed prior to acceptance.

Reading your new additions I noticed some spelling mistakes and differences in font size. Please double-check all new additions, as PeerJ does not provide a full linguistic check prior to publication.

Once again, thank you for submitting your manuscript to PeerJ and I look forward to receiving your revision.

·

Basic reporting

This is a re-review and my previous comments stand.

Experimental design

The author has partly taken on board my comments about transverse stability, but addressed them in a convoluted and unnecessarily complex fashion, which in fact does not quite answer the question. The results are for idealised 2 dimensional "animals" and thus produce results that could questioned in relation to the actual morphology.

In fact, I think the authors conclusions about the instability of the spinosaurids will not change if the whole body stability is calculated, and that actually seems to me to be a very important argument against fully aquatic ability. As such, the paper is OK as it stands, but why not include a more robust stability estimation? For this reason, I think a final revision is required to polish an otherwise fine piece of work.

It really is quite simple to do the calculations for the complete animal, as I will set out below in the comments to the author.

Validity of the findings

The author has addressed my concerns about sensitivity, so no further comment (but see above).

Additional comments

Don, the calculation of the transverse stability of a floating body is relatively straightforward, and gives the answer for the whole body, as distinct from an idealised 2D cross section as in your additions to the paper. The way it works is set out in naval architecture text books, but in brief:

First you calculate the metacentric height, conventionally BM, the distance between the centre of buoyancy and the "metacentre", the point on the centreline where the line of action of the vertical buoyancy force crosses.

BM is simply the second moment of area of the waterplane (I) divided by the displaced volume (V). So:

BM=I/V.

A body is stable if the distance from the centre of gravity to M (GM) is positive.

I have done quick calculations for the alligator (using screen grabs from your paper, so accuracy may not be great) and can send you the spreadsheet if it helps.

Reviewer 2 ·

Basic reporting

The background provided must be improved.

Let me use an example I have already mentioned (but these comments have not really been addressed):

“The radical claim by Ibrahim et al. (2014) of a semi-aquatic theropod dinosaur inspired further investigation of the aquatic potential of Spinosaurus,”

As mentioned in the previous review, why under-cite here? There are a number of papers dealing with this topic, in particular Amiot et al’s landmark work using oxygen isotopes. After all, the title of their paper is "Oxygen isotope evidence for semi-aquatic habits among spinosaurid theropods". That is a very clear statement, right in the title.

The author also needs to address (or at the very least mention) the important and well-documented conclusions these authors reached: "On the basis of the oxygen isotopic composition of their phosphatic remains compared with those of coexisting terrestrial theropod dinosaurs and semiaquatic crocodilians and turtles, we conclude that spinosaurs had semiaquatic lifestyles, i.e., they spent a large part of their daily time in water, like extant crocodilians or hippopotamuses."

Why is this key paper (and others published on potentially semiaquatic dinosaurs over the last few years), published in the high-impact journal Geology, largely ignored in the first few pages?

The author (at the end of the MS) claims that Amiot et al infer fish-eating and not much else. Again, this is clearly not consistent with what the authors actually concluded (see above).

PeerJ guidelines clearly state that literature should be cited appropriately and enough background should be provided. This is not always the case here.

Another example that has not been addressed at all:

“In particular, the hind limbs of the new restoration, although from a single
individual, were not associated with the dorsal axial skeleton which forms a substantial part of the body.”

This seems to contradict Ibrahim et al 2014. Unless the author had access to unpublished data from these authors suggesting otherwise, this statement needs to be changed.

“Again, the forelimbs were not found in association with the hindlimbs , are composites derived from isolated remains, and not associated with the portions of the dorsal axial skeleton.”

Again, this contradicts the paper cited, which states that at least two forelimb elements were associated with the axial elements. If the author did have access to unpublished data contradicting this, it needs to be mentioned here (at the very least a "pers. com.").

These issues must be addressed, considering that they simply relate to accurate referencing and reporting of previously published work.

Experimental design

The author has addressed some of the issues mentioned (though not all - see previous round of comments).

The research questions are well-defined. The methods are mostly described in sufficient detail, and overall this is quite an impressive piece of work, but more information should be provided on the caveats and limitations.

Some details should be explained a little further. Example:

“The sail was assumed to be covered with skin to a depth of 1 cm on both sides”

Not sure where some of these estimates come from - some background would be helpful.

“It was suggested by a reviewer that a test of the software and methods should also be 162 done with a living, aquatic, predatory theropod, ie. a diving bird, to see how it would compare to Spinosaurus. This was done with a model of an emperor penguin (Aptenodytes forsteri Gray 164)."

This is very helpful and really improved this section of the manuscript.

It is important to note somewhere that the work presented here is not based on first-hand study of the fossil material.

Validity of the findings

Not addressing some previously published results (e.g. Amiot et al.), or limitations of the approach taken here, or implying that this paper trumps all other lines of evidence, weakens this contribution.

A more nuanced, balanced reporting of the findings would considerably improve this manuscript.

Additional comments

This is certainly an improved version, but a number of issues need to be addressed before this contribution is consistent with PeerJ guidelines (see detailed comments above). The author is a world-leading expert in his field. Unfortunately, unlike comparable papers published in the past (e.g. Henderson and Naish 2010; Henderson 2006), this particular manuscript is still lacking important context (see above) and a more detailed account of the limitations of the approaches taken here.

I am recommending "minor revisions", even though the list of issues that have not been fully addressed is relatively long (see also first review). At the very least this contribution needs:

1) a more complete and detailed background section, mentioning previous research on spinosaurid lifestyles and addressing other lines of evidence more fully (this is particularly true for the crucial - but under-cited - Amiot et al study, but also other papers, e.g. Gimsa et al. 2016, Vullo et al. 2016). More context on other papers relating to potentially semiaquatic dinosaurs (e.g. Ford and Martin 2010) would also be helpful.

2) a more detailed account of the limitations of this study (this includes methodological weaknesses , as well as issues of data quality (e.g., original fossil material was not examined to build the model)).

3) a possible alternative explanation as to why Spinosaurus shows so many anatomical features typically associated with increased semiaquatic, or aquatic, lifestyles in other vertebrates. Simply dismissing them (as is somewhat implied in the MS) is probably not the right approach. Again, a more nuanced discussion of findings would be very helpful here (e.g. can all those different results be reconciled in some way? If not, what might that mean?). Also, as mentioned in my previous review, if Spinosaurus is “like Baryonyx”, why is its skull and postcranial anatomy so different?

A few questions re: the rebuttal file:

"Comments “Page17”, “Page19”: this is just a petty, defensive detail. No action taken."

Please specify which comments this refers to. This doesn’t seem to match the comments on these pages from my review -

"Comment “Page 8, line 57”: Why not mention the widespread public interest in the new interpretation of Spinosaurus? Does it not indicate the appeal of the subject, demonstrate the engagement of the public with the science, and show why it is worth doing the science?"

Sure, but not in a paper on a completely different subject, i.e. this is not a paper on the public understanding of science or the importance of palaeontological research. I am not sure how exactly this relates to the subject matter of this technical paper.

Finally, the paper ends on the following note:

- Page 20, final part of conclusion: “Lastly, the new reconstruction of Spinosaurus is based on a composition of remains from multiple individuals of varying sizes and proportions that come from different locations, and were scaled to match the presumed proportions of a single individual. This does not seem like a good platform for building hypotheses about what this animal was like as a once living organism.”

This sounds like a rather sweeping criticism of much of palaeontological research (many, many dinosaur palaeontology papers would fall in that category). Is this not what paleontologists have to work with much of the time? Does the building of hypotheses, even when they are built on incomplete skeletons, advance the science of palaeontology? As mentioned before, the author (Henderson, 2010) has used the same approach when reconstructing the weight and flight abilities of the giant pterosaur Quetzalcoatlus, a taxon known from (often very) incomplete remains representing multiple individuals of varying sizes, from different locations in Texas’ Big Bend National Park (Javelina Formation), and frequently scaled to match the presumed proportions of a single individual (see Henderson 2010).

I am not sure the paper needs to end on such a note, but since this is not a technical comment, this is up to the author.

·

Basic reporting

As previous review

Experimental design

As previous review

Validity of the findings

As previous review

Additional comments

I commend the author on this edited and expanded study. The extra inclusions I think work really well! All of my comments have either been satisfactorily addressed, or were merely a difference in opinion regarding reporting style (i.e. inclusion or not of fig. 7) and so are entirely at the authors discretion to follow. I look forward to seeing this work published.

---

## Round 0.3 · Minor Revisions

Dear author,

I have accepted the decision of 'minor revisions' from the two reviewers. They both seem to agree that the MS is converging on a publishable version, but both have given some outstanding issues that need to be addressed before acceptance.

Also, the use of Wikipedia as a reference - can you look into the primary or secondary literature to obtain the body mass data? I assume the relevant wikipedia page (which is not listed), will give you a reference you can look into.

I look forward to receiving you revised manuscript, and thank you for choosing PeerJ.

·

Basic reporting

As before

Experimental design

The author has made an attempt to use the conventional engineering approach to stability calculation, but I am afraid that I still have reservations and comments. I am sorry if this appears pedantic, but I do believe that there are simple ways in which the stability differences between the different the animals can be convincingly demonstrated.

As I have typed my comments as a separate file, please see attached.

Validity of the findings

As before

Additional comments

Please see attached.

Reviewer 2 ·

Basic reporting

MS looks much better. The model still seems to be lacking a few key features (e.g., dense bone in other parts of the skeleton - not just the hind limbs), but these potential weaknesses can be explained in the text.

As already discussed previously, only Gimsa et al. appear to argue in favor of diving and "fish-like" pursuit of prey under water. The starting point and some of the narratives of this MS are, in some ways, based on a strawman "opponent". Most authors have simply advocated "semiaquatic habits/lifestyle", which, according to most definitions, does not necessarily imply deep water hunting/diving etc. Shallow water hunting is not an option that has been ruled out by other authors.

In other words, most of the results presented here do not appear to really contradict previous papers (Gimsa et al being an exception). The MS should definitely reflect this - in its current form, it still doesn't.

Experimental design

The model still seems to be lacking a few key features (e.g., dense bone in other parts of the skeleton - not just the hind limbs), but these potential weaknesses can be explained in the text.

Without a detailed osteohistological analysis of the partial skeleton, the model presented here remains quite speculative (e.g. distribution of dense bone etc.). This should be reflected in the text.

Validity of the findings

The data is not as robust as it could be (see above), and this needs to be reflected in the text.

Additional comments

The unique (amongst dinosaurs) position of the fossa for the fleshy nostril should probably be discussed in more detail. Why would a more–or–less terrestrial animal have such a unique adaptation?

Ibrahim et al. list “Short centra and reduced neural arch articulations” in the tail, features that differ from most other theropods and would presumably have increased flexibility. This should be addressed in a relevant section.

Line 73 – See previous comments. “Media attention” has no relevance to the topic discussed here.

Line 162 – “Lacking evidence to the contrary” is not a strong argument in favor of the selected thickness…

Line 275 – is this Wikipedia reference reliable?

Line 314 – Maybe better to include this in the figure.

Line 329 – Why “caimans” and not “caiman”?

Line 379 – This doesn’t make sense. Why is solid bone assumed to be restricted to just the hind limbs? Many extant aquatic and semi-aquatic vertebrates also have very dense ribs (see manatees for example) etc. – why should Spinosaurus be any different? And what about the dense spines?

Line 391 – There is a word missing here.

Line 529-533 – see previous rounds of comments.

---

## Round 0.4 · Minor Revisions

Dear author,

I have made the decision of 'minor revisions'. Unfortunately, I had to get a new reviewer, as one of the reviewers from the previous rounds was unavailable.

I think the changes needed to be made will be quick and easy to make. And I look forward to receiving your revised version.

·

Basic reporting

This is my third review of the paper, so most comments as before.

Experimental design

The analysis of lateral stability is now acceptable.

Validity of the findings

As before

Reviewer 4 ·

Basic reporting

The paper is well written.

The literature cited is sufficient and a broad background is presented.

The raw data shared are sufficient.

The paper is relevant

Experimental design

The dataset used for testing the hypothesis is adequate and impressive. The data are novel and the new quantification for swimming modes are excellent. However, I have some doubts regarding the material and methods. The model of Spinosaurus is slightly different from the one previously published, and it is not clear from the text what changed and why. Several tests are giving different results from previous papers, but it is not explained in which way the analyses reported in Henderson are different from the previous ones. Osteosclerosis in Spinosaurus is only limited to the appendicular skeleton in this study, even though it was found in the dorsal spines as well. Moreover, osteosclerosis is not reported in some tests. In the end, extant taxa should be used for every test in the paper, in order to validate the results and the interpretation of the analyses.

Validity of the findings

I would like to see more tests to validate the results already obtained in the paper

Additional comments

The paper is relevant and exciting, but needs more tests to back up the claims

Annotated reviews are not available for download in order to protect the identity of reviewers who chose to remain anonymous.

---

## Round 0.5 · accepted · Accept

Dear author,

After reading your rebuttal letter, I am happy to accept your manuscript for publication in PeerJ.

Production staff will get back to you shortly on your proofs.

Once again, thank you for using PeerJ, and I hope you use us again as your publication venue.

#